# TRAJECT-BENCH: A TRAJECTORY-AWARE BENCHMARK FOR EVALUATING AGENTIC TOOL USE

**Pengfei He**[1], **Zhenwei Dai**[2], **Bing He**[2], **Hui Liu**[2], **Xianfeng Tang**[2], **Hanqing Lu**[2], **Juanhui Li**[2],
**Jiayuan Ding**[3], **Subhabrata Mukherjee**[3], **Suhang Wang**[4], **Yue Xing**[1], **Jiliang Tang**[1], **Benoit Dumoulin**[2]
[1]Michigan State University  [2]Amazon Inc.  [3]Hippocratic AI  [4]Penn State University

## ABSTRACT

Large language model (LLM)-based agents increasingly rely on tool use to complete real-world tasks. While existing works evaluate the LLMs' tool use capability, they largely focus on the final answers yet overlook the detailed tool usage trajectory, i.e., whether tools are selected, parameterized, and ordered correctly. We introduce TRAJECT-Bench, a trajectory-aware benchmark to comprehensively evaluate LLMs' tool use capability through diverse tasks with fine-grained evaluation metrics. TRAJECT-Bench pairs high-fidelity, executable tools across practical domains with tasks grounded in production-style APIs, and synthesizes trajectories that vary in breadth (parallel calls) and depth (interdependent chains). Besides final accuracy, TRAJECT-Bench also reports trajectory-level diagnostics, including tool selection and argument correctness, and dependency/order satisfaction. Analyses reveal failure modes such as similar tool confusion and parameter-blind selection, and scaling behavior with tool diversity and trajectory length where the bottleneck of transiting from short to mid-length trajectories is revealed, offering actionable guidance for LLMs' tool use[1].

## 1 INTRODUCTION

Large language models (LLMs) increasingly exhibit strong planning and reasoning abilities (Wei et al., 2022; Yao et al., 2023a): they break goals into subproblems, track intermediate states, and arrange multi-step procedures (Huang et al., 2024; Song et al., 2023). Together, these abilities position LLMs as the "brain" of agentic systems, guiding the systems to perform complex, real-world tasks (Wang et al., 2024; He et al., 2024b; Li et al., 2024). Meanwhile, external tools, such as search engines (Jin et al., 2025), production APIs (Li et al., 2023a), and file/OS operations (Packer et al., 2023), function as the agent's "hands", extending LLM's abilities with precise computations, up-to-date information, and concrete actions. Tool-use has already powered agents across domains: travel agents (Chen et al., 2024; Singh et al., 2024) fuse flight and hotel APIs with visa and weather services to assemble feasible itineraries and resolve constraints; and education agents (Chu et al., 2025; Zhang et al., 2024b) retrieve curriculum materials, generate adaptive exercises, grade against rubrics, and track progress across learning platforms.

Despite these developments, proper evaluations of LLMs' tool-use are essential: they can provide a complete picture of tool-use competence, expose failure modes, and point out concrete directions for improvement. Specifically, the capability to select the right tool(s), determine the correct formats and values of input parameters, conduct multi-step tool utilization, and adapt to unseen tools, makes the core part of evaluating tool-use capability. Benchmarks have been created to test these abilities. For example, Huang et al. (2023) focus on evaluating whether LLMs can determine when to call tools, and Qin et al. (2023); Patil et al. (2024); Patil et al.; Zhuang et al. (2023) evaluate LLMs' capability in calling proper tools for solving complex queries.

However, significant gaps in tool-use evaluation still persist. First, tool-use trajectory complexity is comparatively underexplored: some existing suites rely on small or simulated tools (Zhuang et al., 2023) and many only test short, low-depth tool trajectories (Qin et al., 2023). However, real agents

---

[1]Code available `https://github.com/PengfeiHePower/TRAJECT-Bench`, data available `https://huggingface.co/datasets/bigboss24/TRAJECT-Bench`.

Table 1: Comparison of previous work and `TRAJECT-Bench`.

| | Practical tools | Large&diverse tool | Trajectory structure | Trajectory scaling | Trajectory-aware metrics | Query difficulty | Tool selection | Agentic evaluation |
|---|---|---|---|---|---|---|---|---|
| MetaTool (Huang et al., 2023) | ✓ | ✗ | ✗ | ✗ | ✗ | ✗ | ✗ | ✗ |
| API-Bank (Li et al., 2023a) | ✓ | ✗ | ✗ | ✗ | ✗ | ✗ | ✗ | ✗ |
| ToolBench (Qin et al., 2023) | ✓ | ✓ | ✗ | ✗ | ✗ | ✗ | ✗ | ✓ |
| Gorilla (Patil et al., 2024) | ✓ | ✓ | ✗ | ✗ | ✗ | ✗ | ✓ | ✗ |
| BFCL (Patil et al.) | ✓ | ✓ | ✗ | ✗ | ✗ | ✗ | ✗ | ✗ |
| ToolQA (Zhuang et al., 2023) | ✗ | ✗ | ✗ | ✗ | ✗ | ✓ | ✗ | ✓ |
| TRAJECT-Bench (ours) | ✓ | ✓ | ✓ | ✓ | ✓ | ✓ | ✓ | ✓ |

can possess large tool sets and face complex user queries, so evaluation on a larger executable tool set and trajectories with more tools involved is still needed. Second, the complexity of real user queries is also underrepresented: existing benchmarks often consider straightforward prompts by including the API name directly in the prompt. In contrast, real-world agents may face user queries that consist of indirect language and implicit cues, and the agents need to infer both the choice of the tool and how to set the inputs. Third, most benchmarks still privilege final-answer metrics. For example, Qin et al. (2023) only provide pass rate and win rate of the final answer, and Patil et al. heavily rely on the overall accuracy. In this case, it is hard to track the root cause of the incorrect final answer, which can be caused by various issues such as incorrect tool selection, disorders of tool-use, or incorrect parameterization. This oversight can obscure the evaluation of LLMs' tool-use capability and cannot differentiate it from other general reasoning capabilities, especially given the observation in (Roberts et al., 2020; Qian et al., 2025b) that LLMs can solve the problem using internal knowledge even when the wrong tools are called.

To bridge these gaps, we present `TRAJECT-Bench`–a benchmark to comprehensively evaluate the LLMs tool-use capability via providing **(1) tool-use trajectories** of different complexities and **(2) user queries of different levels of difficulties** given the same tool-use trajectory. Meanwhile, we also include **(3) evaluation metrics** to evaluate the LLMs' tool-use capability from diverse perspectives. As summarized in Table 1, we are the first to provide all the comprehensive analysis perspectives for tool-use evaluation compared to existing benchmarks.

During the data construction, to ensure the quality of the data and better align it with real scenarios, on the tool side, we select a diverse suite of over 1,000 high-fidelity tools drawn from various real-world domains (e.g., finance, travel, music, etc.), exposed via production-style APIs. Given these tools, to construct **(1) tool-use trajectories**, we further synthesize task-driven tool-use trajectories that encode different trajectory structures and scales. We consider both parallel and sequential tool calling trajectory structures and a trajectory scale of 3 to 10+ tool counts. Moreover, for each trajectory, we provide **(2) two semantically aligned queries of different query difficulties**: a direct and explicit "simple" version and a naturalistic and indirect "hard" version. Such a data construction procedure allows us to decompose the queries' difficulty into different perspectives and understand the corresponding specific weaknesses of LLMs.

Based on this data, we evaluate the state-of-the-art models, compare representative tool-selection strategies, and assess agentic tool-use settings beyond individual LLMs. For **(3) the evaluation metrics**, we report trajectory-aware metrics alongside final-answer accuracy: Trajectory Exact-Match and Trajectory Inclusion (whether the required tools are invoked and in the correct order), Tool-Usage (schema constraints, formats, and value checks of the tool's inputs), and an LLM-judge Trajectory-Satisfy score when gold traces are unavailable. Together, these evaluations treat tool-use as a primary skill, with trajectory-aware metrics that explicitly measure trajectory and query complexity. Our key contributions are summarized as follows:

- **Dataset and tasks.** We introduce high-quality, executable tool suites across various domains and tasks. We model trajectory complexities via different structures (*parallel* and *sequential*) and various tool counts. Queries are aligned with trajectories with different difficulty levels.

- **Evaluation suite.** We evaluate state-of-the-art LLMs and representative tool selection strategies. We also include agentic tool-use in the evaluation suite. Beyond final-task accuracy, we introduce trajectory-aware metrics to better capture tool-use capability.

- **Insights and guidance.** We provide analyses based on our evaluation, and reveal important insights, such as models' struggling to infer correct tools from indirect queries, the bottleneck of the transition from short to mid-length trajectories, limitations of retrieval-based selection. These insights can inspire future development of a more precise and reliable tool-use mechanism.

## 2 RELATED WORK

**LLM Agents**. LLM-based agents treat large language models as the "brain" that plans, acts, and reflects while interfacing with external tools and environments (Wang et al., 2024; Guo et al., 2024). Early formulations follow a think–act loop: ReAct interleaves chain-of-thought with grounded actions (Yao et al., 2023b; Wei et al., 2022), and Reflexion adds self-critique and memory to correct future behavior (Shinn et al., 2023). Building on this, tool-augmented frameworks formalize function calling, tool selection, and state management to operate retrieval, code execution, and other systems (Schick et al., 2023; Lewis et al., 2020). These designs have made LLM agents effective and flexible across domains (He et al., 2024b; Trivedi et al., 2022; Li et al., 2023b). Web agents perceive pages and perform UI-level actions for tasks like shopping and booking (Yao et al., 2022; Zhou et al., 2023; He et al., 2024a; Zheng et al., 2024); code agents iteratively edit, test, and debug multi-file projects (Yang et al., 2024; Hong et al., 2024); embodied agents couple language with perception and control (Kim et al., 2024; Mao et al., 2023; Zhang et al., 2024a); and scientific agents use tools and simulation for literature review, hypothesis generation, and experiment planning (Park et al., 2023; Ren et al., 2025).

**Tool-use in LLM agents**. Tool use is a core capability of LLM agents and has drawn substantial interest (Qu et al., 2025; Qin et al., 2024). Methods advance when and how models invoke functions: RL approaches (ReTool, ToolRL, OTC) optimize selection and calling (Feng et al., 2025; Qian et al., 2025a; Wang et al., 2025); supervised/instruction tuning (e.g., Toolformer; small-model instruction tuning) teaches adherence to schemas (Schick et al., 2023; Shen et al., 2024); and feedback frameworks (TRICE) add self-correction (Qiao et al., 2023). Complementary evaluations measure these abilities: Gorilla grounds calls in public APIs (Patil et al., 2024); the Berkeley Function-Calling Leaderboard scores cross-domain execution (Patil et al.); ToolBench scales to multi-step RapidAPI tools (Qin et al., 2023); ToolQA targets tool-augmented reasoning (Zhuang et al., 2023); and Meta-Tool probes tool awareness and selection (Huang et al., 2023). Together, this work advances both capability and measurement for tool-using agents. However, existing benchmarks seldom treat tool use as the primary objective. Most prioritize end-task scores and overlook trajectory-level signals, limiting the insight into agent behavior and opportunities for improvement. Therefore, a benchmark with a high-fidelity, diverse tool set and realistic, task-driven queries spanning difficulty levels, enabling trajectory-aware evaluation and diagnosis, is still urgently needed.

## 3 DATA CONSTRUCTION

To construct the data, we first select the candidate tools to be included in the benchmark (Section 3.1), and then curate ground truth tool-using trajectories and corresponding queries (Section 3.2).

### 3.1 TOOL SET CURATION

Following (Qin et al., 2023), we source practical tools from RapidAPI, which contains many APIs for real-world tasks. To keep the effort tractable, we curate data for ten representative domains: travel, mapping, finance, weather, e-commerce, news/media, gaming, email, education, and music, where LLM agents are especially popular and useful[2].

Although thousands of APIs are available from RapidAPI, many of them contain unclear descriptions or parameters, and some are even non-executable. Therefore, we carefully select and revise the APIs based on the following four requirements.

(1) **Executable tools with meaningful outputs**. We validate each tool by executing it across parameter combinations and discard those that error out (details in the Appendix A.1). For the remainder, we use an LLM to summarize outputs and formats, removing tools whose outputs are semantically trivial, that is, do not contribute useful information.

(2) **Clear, action-oriented tool descriptions**. Because many APIs ship with sparse or vague documentation, we refine API descriptions by combining the original description with empirical I/O observed during validation. For instance, the original description is just 'Get price (symbol),' yet empirical calls reveal it enforces a 50-item page_size cap and returns {price, currency, timestamp}; we merge these behaviors into the clarified description.

(3) **Minimal functional overlap**. We deduplicate identical tools/APIs (e.g., multiple flight-search endpoints) to avoid ambiguity and ensure deterministic trajectory evaluation, keeping a single repre-

---

[2]More domains can be added following our data-generation pipeline

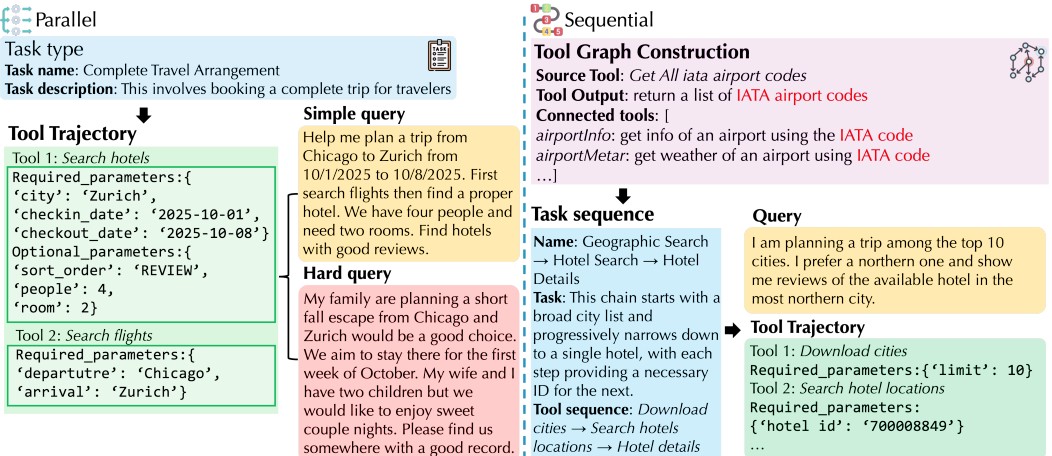

Figure 1: An illustration of data in `TRAJECT-Bench`. The left side demonstrates the parallel query where tool trajectories are created based on real task types and then queries with two difficulty level are generated. The right side shows the generation process of sequential queries, where a tool graph is first built, then task sequences are manually designed and finally detailed queries and trajectories are created.

sentative per function. Closely related tools are kept when they introduce distinct parameterizations to increase task complexity. This is done with the help of LLMs, followed by manual verification.

(4) **Controlled tool complexity**. We manually keep tools with strong parameter complexity (number of fields, types, and constraints) and remove some with simpler parameters (e.g., no inputs needed), ensuring the set contains rich tools to stress-test tool-use competence. Finally, we obtain a high-fidelity tool set $\mathcal{T}$. Details and examples of the curated tool set can be found in the Appendix A.1.

## 3.2 TASK-DRIVEN QUERY GENERATION

To evaluate LLMs' tool use in realistic settings while keeping the evaluation controllable, we synthesize queries from real-world *task types*, which denote categories/families of tasks that users can adopt the agent for, such as booking flights and arranging stays in the destination (without details of dates or locations). Specifically, as shown in the left side of Figure 1, for each domain we manually collect representative task types (e.g., "real-time trip monitoring & assistance" for travel; "creating math & science study materials" for education), and provide a precise description for each task type. For every task type, we create tool-use trajectories with details and pair them with aligned user queries. By templating from task types, we minimize human efforts in collecting and annotating data, yet keep queries faithful to real use cases. This also enables diverse, controllable, and scalable query–trajectory data across domains.

We consider two basic trajectory structures: (i) *parallel*, where tools operate independently (the choice/execution of one tool does not depend on others); and (ii) *sequential*, where tools form a chain and later steps depend on earlier outputs[3]. Details of trajectory and query construction are provided as follows.

### 3.2.1 PARALLEL QUERY

**Trajectory generation**. For each domain, we prompt an LLM to synthesize valid, logical tool-using trajectories from a task-type description and the domain's available tools. To keep evaluation scalable and comparable, we enforce two rules: (i) the plan must use a specified number of tools (typically 3–10+); and (ii) tools are presented as a parallel-ready set—each call is self-contained (inputs fixed up front) and independent of the others. Parallel trajectories are encoded as unordered sets of tool calls with fully specified inputs. For example, Figure 1 shows that combining hotel and flight search APIs enables end-to-end travel planning.

**Query generation**. After trajectories are finalized, we pair each trajectory with user queries designed at two difficulty levels:

---

[3]We focus on parallel and chain structures to balance data cost and evaluation reliability. We present experiments on hybrid structures in Appendix C, and leave richer graph topologies for future work.

- **Simple version**. Provide a straightforward and precise instruction that explicitly specifies the need for tools and their key parameters, such as the example in Figure 1 where the simple query gives direct and detailed requests for a travel plan.

- **Hard version**. Present a more challenging, indirect request that conveys the same constraints via natural cues and implications rather than explicit instructions. This mirrors real interactions where users state goals colloquially (e.g., *"hotels with good record"* rather than *"sort hotels by review score"* as shown in Figure 1).

After the generation, both trajectories and queries go over LLM-based automatic validation and human inspection to ensure the quality and reduce ambiguity. By jointly varying (i) the number of tools per trajectory and (ii) the difficulty level of paired queries, we obtain a comprehensive yet well-controlled evaluation of LLMs' tool-use capabilities.

### 3.2.2 SEQUENTIAL QUERY

Sequential queries require strong dependencies between tools within the trajectory, which makes it difficult for an LLM to directly propose a correct chain end-to-end. To address this, we first build a **directed tool graph** $G_T = (V, E)$ that captures how tools can feed one another: each tool $t$ in $V$ is a node, and we add a directed edge $t_1 \to t_2$ to the edge set E, when the information from the output of $t_1$ can be used as input parameters of $t_2$. For instance, as shown in the right side of Figure 1, the IATA codes returned from the GetAllIATA API can be used as the input for airportInfo API to retrieve detailed information of a specific airport, so we connect them.

Given $G_T$, we generate representative tasks and corresponding tool sequences with a controlled number of tools (e.g., 5 sequences for each number of tools). Unlike the parallel setting, these sequences are manually created as trajectory templates to ensure logical coherence, e.g., $t_1 \to t_2 \to \cdots \to t_{n_traj}$, where $n_t raj$ denotes the number of tools involved. Each template explicitly specifies the parameter bindings between adjacent tools, i.e., which output results from $t_i$ populate which input fields of $t_{i+1}$. For instance, the task sequence in Figure 1 describes a general 'city-¿hotel location-¿hotel detail' chain without details such as city name or hotel id.

Similar to the generation of parallel trajectories, we then prompt an LLM to create trajectories with details and corresponding user queries from each template (5 per template), yielding $s_{traj}^{seq} = \{t_1(params) \to \cdots \to t_{n_{traj}}(params)\}$ together with $q_{traj}^{seq}$, as shown in Figure 1. All trajectories and their paired queries undergo automated validation and human review to ensure validity, task alignment, and overall quality. More details of the generation process and datasets are shown in the Appendix A.2. This pipeline produces coherent tool chains and tightly aligned queries at varied depths (number of tools involved), enabling scalable and transparent evaluation.

Finally, the whole dataset contains 1228 tools and 5670 queries (details in Appendix A.2).

## 4 EVALUATIONS ON TRAJECT-BENCH

We comprehensively evaluate LLMs' agentic tool-use on TRAJECT-Bench, focusing on key research questions that highlight its benefits and potential:

**RQ1.** When and why do LLMs succeed or fail at tool-use?
**RQ2.** Can retrieval-augmented selection improve tool selection and parametrization?
**RQ3.** Do agentic methods, including training and inference, improve tool-use capability?

### 4.1 EVALUATION SETTINGS

**Test (query) methods.** When evaluating LLM's own capability in calling proper tools to solve queries, we mainly adopt the direct query where user queries and tools are provided to the model. We provide additional results using Chain-of Thought in Appendix C.

**Tool selection strategy.** Unlike existing benchmarks (Qin et al., 2023; Huang et al., 2023) ship only retrieved tools (not the full tool set) during evaluation, we treat the tool selection strategy as a crucial part of tool-using, especially when the tool set is large and exceeds the model's context window. We evaluate three common tool selection strategies (labels shown in Table 4): all—provide the full tool set in context; domain—provide only tools from the query's domain; and retrieval—retrieve a subset based on the query and tool descriptions. For retrieval, we test two widely used embedding models,

all-MiniLM-L6-v2 (Solatorio, 2024) and bge-large-en-v1.5 (Xiao et al., 2023), plus a tool-specific retriever, ToolBench-IR (Qin et al., 2023). Unless noted, we retrieve 20 tools by default.

**Test models.** We test on a wide range of state-of-the-art LLMs: Claude family, including Claude-3.7 (ant, 2024) and Claude-4 (ant, 2025); Gemini family, including Gemini-2.5-pro and Gemini-2.5-flash; GPT family, including o4-mini and gpt-oss-120B; Qwen3-235b-A22B, the latest generation of LLMs in the Qwen series; DeepSeek-V3.1, the latest DeepSeek model; and Kimi-k2, the latest MoE model from MoonShot AI.

**Agentic evaluation.** Despite the tool-using capability of models themselves, one highlight of our benchmark is the evaluation of models' agentic tool-using capability. We consider two agentic capabilities [4]. The first one is the capability specifically trained in the models. The SOTA LLMs, such as Claude, Gemini, Deepseek, and Kimi-k2, all include tool-using in the training and directly support tool-calling, e.g., the "tools" input in `client.messages.create` of Claude APIs, and do not need to additionally list them in the context of the prompt. Therefore, we provide evaluations on these models' internal tool-using capability and compare with the results of providing tools as context. More details can be found in Appendix B.

We also evaluate LLM agents for tool-use. Given diverse designs, we focus on ReAct (Yao et al., 2023b)—a foundational blueprint for many real agents (e.g., MetaGPT (Hong et al., 2024)). We pair ReAct with domain-specific and retrieval-based tool selection. Because ReAct reasons and executes over multiple turns, we test two retrieval modes: *static* (retrieve once from the query) and *dynamic* (retrieve at each reasoning/execution turn).

**Metrics.** Unlike existing benchmarks, which mainly focus on the final performance, we consider two categories of metrics: trajectory-aware metrics and final performance metrics. For the trajectory-aware metrics, we consider (1) Exact match (EM), which compares the predicted tool-using trajectory and the ground-truth ones to check if the predicted tools (names, not parameters) are exactly the same as the ground truth ones; (2) Inclusion, which measures what proportion of ground truth tools is included in the predicted tool trajectory; (3) Tool Usage (Usage), which checks if the predicted tool parameters match the ground truth ones; and (4) trajectory satisfaction (Traj-satisfy), where we prompt an LLM judge (Claude-4 by default [5]) to determine to what extent a predicted trajectory can solve the user query, and by using this metric we mimic a real-world scenario when the ground truth is not available. For the final performance metric, we report (5) Acc, which measures if the predicted final answer matches the ground truth answer by prompting an LLM judge. For the retrieval-based methods, we include the (6) retrieval rate, which measures what proportion of ground truth tools are retrieved. More details can be found in Appendix B.

## 4.2 RQ1. WHEN AND WHY DO LLMs SUCCEED OR FAIL AT TOOL-USE?

We first evaluate the tool-use capability of individual LLMs. For each evaluating query, we feed the LLM both the query and the available tools as the context. Since the full tool set is beyond the length of the model's context window, we focus on tools related to the same domain of the query, e.g., we only provide Travel tools for a query in the Travel domain. We report metrics mentioned in Section 4.1 in Table 2 for parallel data and Table 3 for sequential data.

**Overall tool-use performance analysis.** Generally, we observe that Gemini-2.5-pro performs best on simple queries, while Claude-4 ties with it on the hard versions. Less capable models, such as Claude-3.7 and Gemini-2.5-flash lag behind. The LLM-judge Traj-Satisfy score also tracks EM closely (e.g., Claude-4: $8.549 \leftrightarrow 0.846$ on simple; $4.882 \leftrightarrow 0.445$ on hard), indicating the judge is an effective proxy for EM.

In addition, we make the following key observations. (a) Most models perform well on the simple versions but struggle on the hard versions. For most metrics on most models, we observe a clear gap between simple and hard; for example, EM for Claude-4 drops from $0.846 \rightarrow 0.445$ and for Gemini-2.5-pro from $0.851 \rightarrow 0.442$ under direct prompting. This suggests that when tool choices and constraints must be inferred from indirect cues, models often miss the exact tool set and parameters, and future efforts should focus more on these more complex cases. (b) For nearly every

---

[4]We note that agent structures are very diverse, and we include additional experiments in Appendix C. We will keep exploring other agents and extending the `TRAJECT-Bench`.

[5]This evaluation does not require advanced models, and we also leverage smaller and open-source models in Appendix C

model, *parallel–simple* obtains better performance than *sequential*. For example, the EM, Inclusion, and Usage of Gemini-2.5-pro $(0.851, 0.854, 0.835)$ on simple parallel queries are higher than those on sequential queries $(0.807, 0.821, 0.809)$, suggesting that inter-step dependencies and ordering introduce additional challenges for both tool selection and parameter determination. This motivates future tool-use training and inference methods that explicitly model dependency and order. (c) Inclusion typically exceeds EM, especially on hard parallel and for weaker models. For most hard cases, there is a pronounced Inclusion–EM gap—e.g., $0.135$ vs. $0.554$ for Claude-3.7 and $0.216$ vs. $0.538$ for Gemini-2.5-flash—indicating that models fail to recover the complete set.

**Tool-use scaling analysis**. Since the user queries can be very complex in practice and can involve many tools, we analyze the model's tool-use capability when the number of tools in the (ground truth) trajectory increases. As an illustration, we present the results on both simple and hard queries on models from different families in Figure 2, where the x-axis is the number of tools and the y-axis is the metric EM. As the number of tools in a trajectory grows, all models experience a clear decline in performance, with the steepest drop occurring between three and five tools. Among the evaluated models, Claude-4 and DeepSeek show the strongest generalization, maintaining relatively stable performance even at longer tool chains, whereas o4-mini and Kimi-k2 collapse sharply beyond seven tools. These results reveal that **scaling tool use is a universal challenge**, where **the main bottleneck lies in the transition from short to mid-length trajectories**, and underline the importance of improving models' long-horizon tool-use and error recovery capabilities.

**Deeper analysis on tool-use failures**. We further take a look at failed cases to identify common failure patterns in tool use, and detailed examples are presented in Appendix D. (a) **Similar tool confusion**. This refers to the cases when models face tools that are partially overlapping capabilities but distinct scopes, inputs/outputs, or constraints (e.g., confusing between Spotify: Search vs. YouTube Music: Search). Distinguishing among these tools requires a more delicate understanding of the tools' exact functionalities, parameters, outputs, etc. Existing models need more improvement to obtain a precise tool match facing such distractions. (b) **Parameter-blind tool selection & usage**. In some cases, models overlook tool parameters (values, formats) when selecting tools, relying primarily on tool descriptions. This can lead to selecting tools with wrong parameters,cascading failures in downstream steps, and degraded end-task accuracy. We also notice that this failure is much rarer for Kimi-k2, suggesting that delicate agentic tool training can effectively improve this problem. (c) **Redundant tool calling**. This refers to cases where the model invokes more tools than necessary. We observe two forms of redundancy: (i) *related-but-unhelpful calls*—e.g., for a query about airport information in Zurich, the model additionally calls "Get All IATA airport codes," which returns every code irrespective of the city name provided in the query; and (ii) *unrelated calls*—e.g., invoking railway tools when the task asks only for flight information. The first pattern likely reflects a conservative "cover-all-bases" strategy, whereas the second is more consistent with hallucination or weak intent grounding. Since redundant calls can cause inflated latency and cost and even errors, precise tool-calling trajectory data should be included in training for improvement, and careful validation can also be adopted. (d) **Struggle to infer intents from hard queries**. When facing indirect queries, models often misinterpret the user's intent, leading to entirely irrelevant tool selections.

Together, these failure patterns show that models struggle with accurate tool choice and parameter use in complex scenarios, underscoring the need for better training and inference strategies.

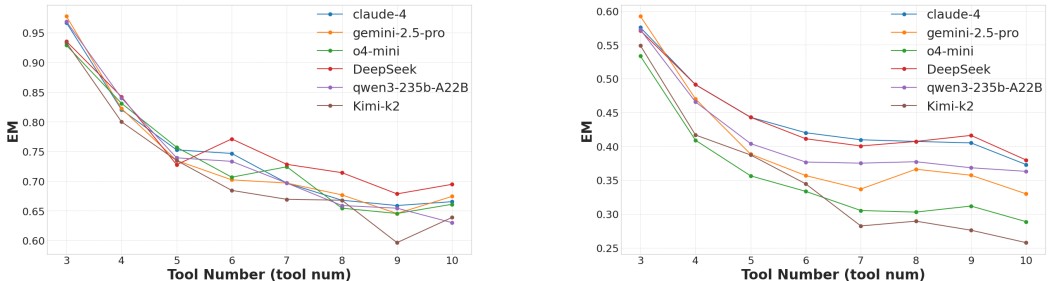

Figure 2: Figures illustrating model's scaling tool-use behavior. The x-axis denote the number of tools in the trajectory and y-axis denotes the metric EM. Left is for simple queries, while right is for hard queries.

Table 2: Results of individual model's tool-use capability with domain-specific tools on parallel queries.

| Model | Simple | | | | | Hard | | | | |
|---|---|---|---|---|---|---|---|---|---|---|
| | EM | Inclusion | Usage | Traj-Satisfy | Acc | EM | Inclusion | Usage | Traj-Satisfy | Acc |
| Claude-3.7 | 0.676 | 0.746 | 0.706 | 6.831 | 0.714 | 0.135 | 0.554 | 0.603 | 1.582 | 0.246 |
| Claude-4 | 0.846 | 0.855 | 0.839 | 8.549 | 0.905 | 0.445 | 0.668 | 0.794 | 4.882 | 0.517 |
| Gemini-2.5-flash | 0.714 | 0.759 | 0.784 | 7.215 | 0.782 | 0.216 | 0.538 | 0.666 | 2.340 | 0.263 |
| Gemini-2.5-pro | 0.851 | 0.854 | 0.835 | 8.599 | 0.911 | 0.442 | 0.652 | 0.785 | 4.849 | 0.498 |
| GPT5-mini | 0.817 | 0.825 | 0.819 | 8.236 | 0.834 | 0.155 | 0.589 | 0.626 | 1.701 | 0.380 |
| o4-mini | 0.823 | 0.830 | 0.824 | 8.316 | 0.863 | 0.376 | 0.629 | 0.726 | 3.849 | 0.472 |
| gpt-oss:120b | 0.652 | 0.667 | 0.715 | 6.588 | 0.726 | 0.174 | 0.559 | 0.671 | 1.909 | 0.387 |
| DeepSeek | 0.833 | 0.836 | 0.829 | 8.417 | 0.889 | 0.439 | 0.669 | 0.757 | 4.817 | 0.458 |
| qwen3-235b-A22B | 0.844 | 0.856 | 0.809 | 8.529 | 0.898 | 0.440 | 0.667 | 0.796 | 4.828 | 0.479 |
| Kimi-k2 | 0.815 | 0.876 | 0.872 | 8.236 | 0.902 | 0.321 | 0.666 | 0.772 | 3.522 | 0.448 |

Table 3: Results of individual model's tool-use capability with domain-specific tools on sequential queries.

| Model | EM | Inclusion | Usage | Traj-Satisfy | Acc |
|---|---|---|---|---|---|
| Claude-3.7 | 0.583 | 0.724 | 0.584 | 6.010 | 0.573 |
| Claude-4 | 0.819 | 0.832 | 0.775 | 8.243 | 0.813 |
| Gemini-2.5-flash | 0.613 | 0.695 | 0.714 | 6.219 | 0.652 |
| Gemini-2.5-pro | 0.807 | 0.821 | 0.809 | 8.119 | 0.848 |
| GPT5-mini | 0.693 | 0.715 | 0.692 | 7.042 | 0.677 |
| o4-mini | 0.789 | 0.807 | 0.748 | 8.134 | 0.761 |
| gpt-oss:120b | 0.538 | 0.619 | 0.694 | 5.546 | 0.653 |
| DeepSeek | 0.825 | 0.849 | 0.811 | 8.305 | 0.823 |
| qwen3-235b-A22B | 0.824 | 0.837 | 0.772 | 8.194 | 0.791 |
| Kimi-k2 | 0.821 | 0.827 | 0.793 | 8.363 | 0.833 |

## 4.3 RQ2. CAN RETRIEVAL-AUGMENTED SELECTION IMPROVE TOOL SELECTION AND PARAMETRIZATION?

Since tool sets can be large and diverse in practice, it is preferable to first narrow them to a smaller, more relevant subset to avoid adding irrelevant tools to the context. Retrieval-based strategy is a popular way to achieve this, and we evaluate different embedding models and different retrieval pools in experiments to find out to what extent retrieval can help. The results in Table 4 suggest two clear findings. (a) **When the retrieval pool is already restricted to domain-related tools, retrieval adds little benefit for simple queries**. For both Claude-3.7 and Claude-4, EM and accuracy remain nearly the same across embedding models, with ToolBench-IR showing only marginal improvements. (b) **Retrieval becomes a severe bottleneck for hard queries**. We notice that the retrieval rate significantly drops (merely over 50% for most models) and all the performance metrics drop sharply compared with the non-retrieval cases. The core issue may be that retrievers heavily rely on semantic similarities and fail to capture underlying intents and steps from implicit queries. Therefore, they struggle to correctly identify necessary tools, causing cascading failures in tool selection and parameterization. Together, these findings reveal limitations of retrieval-based tool selection methods and call for a better strategy.

Table 4: Results of individual models combined with retrieval-based tool-selection strategy.

| Claude-3.7 | | Simple | | | | | | Hard | | | | | |
|---|---|---|---|---|---|---|---|---|---|---|---|---|---|
| Emb model | Retrieval pool | Retrieval rate | EM | Inclusion | Usage | Traj-Satisfy | Acc | Retrieval rate | EM | Inclusion | Usage | Traj-Satisfy | Acc |
| bge-large | Domain | 0.906 | 0.681 | 0.792 | 0.738 | 7.134 | 0.708 | 0.585 | 0.035 | 0.410 | 0.692 | 0.541 | 0.127 |
| | All | 0.842 | 0.639 | 0.762 | 0.728 | 6.592 | 0.665 | 0.482 | 0.020 | 0.341 | 0.657 | 0.334 | 0.098 |
| all-MiniLM | Domain | 0.913 | 0.685 | 0.793 | 0.749 | 7.257 | 0.717 | 0.584 | 0.029 | 0.403 | 0.683 | 0.265 | 0.109 |
| | All | 0.868 | 0.645 | 0.751 | 0.745 | 6.613 | 0.680 | 0.460 | 0.012 | 0.403 | 0.641 | 0.140 | 0.082 |
| ToolLM-IR | Domain | 0.945 | 0.703 | 0.814 | 0.778 | 7.142 | 0.715 | 0.578 | 0.030 | 0.419 | 0.698 | 0.294 | 0.139 |
| | All | 0.877 | 0.652 | 0.783 | 0.758 | 6.770 | 0.696 | 0.475 | 0.024 | 0.425 | 0.656 | 0.259 | 0.132 |
| **Claude-4** | | **Simple** | | | | | | **Hard** | | | | | |
| bge-large | Domain | 0.906 | 0.852 | 0.867 | 0.835 | 8.631 | 0.902 | 0.585 | 0.031 | 0.397 | 0.672 | 0.328 | 0.264 |
| | All | 0.842 | 0.785 | 0.823 | 0.770 | 8.053 | 0.876 | 0.482 | 0.012 | 0.292 | 0.656 | 0.111 | 0.189 |
| all-MiniLM | Domain | 0.913 | 0.833 | 0.859 | 0.786 | 8.764 | 0.870 | 0.584 | 0.033 | 0.410 | 0.663 | 0.355 | 0.277 |
| | All | 0.868 | 0.817 | 0.826 | 0.773 | 8.352 | 0.832 | 0.460 | 0.015 | 0.267 | 0.620 | 0.171 | 0.168 |
| ToolLM-IR | Domain | 0.945 | 0.906 | 0.928 | 0.833 | 9.117 | 0.916 | 0.578 | 0.028 | 0.420 | 0.680 | 0.286 | 0.241 |
| | All | 0.877 | 0.852 | 0.861 | 0.764 | 8.613 | 0.879 | 0.475 | 0.014 | 0.298 | 0.653 | 0.190 | 0.164 |

Table 5: Evaluation of model's inherent agentic tool-use capability, denoted as "agentic". "context" denotes the results providing tools as context, same with Table 2.

| model | | Simple | | | | | Hard | | | | |
|---|---|---|---|---|---|---|---|---|---|---|---|
| | | EM | Inclusion | Usage | Traj-Satisfy | Acc | EM | Inclusion | Usage | Traj-Satisfy | Acc |
| Claude-4 | Agentic | 0.832 | 0.868 | 0.816 | 8.407 | 0.893 | 0.440 | 0.637 | 0.751 | 4.828 | 0.486 |
| | Context | 0.846 | 0.855 | 0.839 | 8.549 | 0.905 | 0.445 | 0.668 | 0.794 | 4.882 | 0.517 |
| Gemini-2.5-pro | Agentic | 0.828 | 0.866 | 0.876 | 8.367 | 0.917 | 0.416 | 0.674 | 0.768 | 4.564 | 0.503 |
| | Context | 0.851 | 0.854 | 0.835 | 8.599 | 0.911 | 0.442 | 0.652 | 0.785 | 4.849 | 0.498 |
| Deepseek | Agentic | 0.819 | 0.857 | 0.756 | 8.256 | 0.845 | 0.416 | 0.682 | 0.783 | 4.564 | 0.481 |
| | Context | 0.833 | 0.836 | 0.729 | 8.417 | 0.889 | 0.439 | 0.669 | 0.757 | 4.817 | 0.458 |
| Kimi-k2 | Agentic | 0.853 | 0.977 | 0.893 | 8.620 | 0.951 | 0.315 | 0.717 | 0.765 | 3.456 | 0.437 |
| | Context | 0.815 | 0.876 | 0.872 | 8.236 | 0.902 | 0.321 | 0.666 | 0.772 | 3.522 | 0.448 |

### 4.4 RQ3. DO AGENTIC METHODS, INCLUDING TRAINING AND INFERENCE, IMPROVE TOOL-USE CAPABILITY?

Despite the individual model's capability of using tool information as the context, agentic methods are also developed in existing literature, including two major categories: one is the agentic tool-use training undergone by most SOTA models (Team et al., 2025; Yang et al., 2025), and the other is to design LLM agents to adopt tools (Yao et al., 2023b). For the first category, we evaluate four representative models that inherently support tool-use, and show results in Table 5, where we also provide results of context-based evaluation as a baseline. Table 5 shows that the agentic tool-use capability is similar to the context-based baseline for most models on both simple and hard queries.

For the second category, we evaluate Claude models on both parallel and sequential queries by combining retrieval with ReAct. We consider two settings: a static mode, where a subset of tools is retrieved once based on the user query, and a dynamic mode, where retrieval occurs before each thought and action. ToolLM-IR is used as the retrieval backbone in all experiments. The results in Tables 6 and 7 show that ReAct consistently improves tool-use performance compared with individual models in Table 2. For example, on parallel hard queries, Claude-4 improves from 0.445 EM (Table 2) to 0.463 EM with ReAct (Table 6), and the performance is further boosted via dynamic retrieval to 0.473 EM. Similarly, Claude-3.7 gains from 0.135 EM to 0.186 EM with domain tools and 0.296 EM under dynamic ReAct. These show that **iterative calling tools based on execution results offer a stronger basis for accurate tool retrieval and usage**.

Overall, these findings suggest that agentic tool learning and inference serve as an effective and robust way to improve a model's tool-use capability.

Table 6: ReAct results on parallel queries combined with different retrieval strategy.

| Model | retrieval mode | Simple | | | | | Hard | | | | |
|---|---|---|---|---|---|---|---|---|---|---|---|
| | | EM | Inclusion | Usage | Traj-Satisfy | Acc | EM | Inclusion | Usage | Traj-Satisfy | Acc |
| claude-3.7 | Domain tool | 0.735 | 0.816 | 0.752 | 7.630 | 0.782 | 0.186 | 0.612 | 0.641 | 1.930 | 0.307 |
| | Static | 0.762 | 0.841 | 0.791 | 7.840 | 0.803 | 0.093 | 0.432 | 0.697 | 1.180 | 0.148 |
| | Dynamic | 0.814 | 0.876 | 0.805 | 8.220 | 0.831 | 0.296 | 0.581 | 0.703 | 3.080 | 0.346 |
| claude-4 | Domain tool | 0.892 | 0.915 | 0.864 | 9.160 | 0.928 | 0.463 | 0.670 | 0.797 | 4.804 | 0.418 |
| | Static | 0.886 | 0.912 | 0.872 | 9.116 | 0.916 | 0.031 | 0.424 | 0.668 | 0.403 | 0.139 |
| | Dynamic | 0.933 | 0.951 | 0.875 | 9.422 | 0.948 | 0.472 | 0.703 | 0.736 | 4.811 | 0.438 |

Table 7: ReAct results on sequential queries combined with different retrieval strategy.

| Sequential | Retrieval Mode | EM | Inclusion | Usage | Traj-Satisfy | Acc |
|---|---|---|---|---|---|---|
| claude-3.7 | Domain tool | 0.615 | 0.735 | 0.625 | 6.384 | 0.634 |
| | Static | 0.609 | 0.692 | 0.644 | 6.266 | 0.603 |
| | Dynamic | 0.651 | 0.749 | 0.651 | 6.574 | 0.634 |
| claude-4 | Domain tool | 0.827 | 0.846 | 0.829 | 8.492 | 0.842 |
| | Static | 0.817 | 0.833 | 0.825 | 8.059 | 0.811 |
| | Dynamic | 0.849 | 0.881 | 0.833 | 8.573 | 0.917 |

## 5 CONCLUSION

This work introduces a comprehensive tool-use benchmark that focuses on the tool-use trajectory. By constructing a large and diverse executable tool suite, modeling trajectories of varying structures

and scales, and pairing them with user queries of different difficulty levels, we provide a realistic and rigorous setting for evaluation. Our trajectory-aware metrics go beyond final-answer accuracy, enabling a clearer understanding of where and why LLMs succeed or fail in tool use. Through comprehensive evaluation and analysis, our benchmark not only highlights the current strengths and limitations of state-of-the-art models but also offers actionable insights for improving agentic tool-use capabilities. We hope this benchmark establishes a foundation for systematic progress in developing LLMs that can reliably plan, select, and execute tools in complex, real-world scenarios. We will also keep including more diverse domains, agent structures, and extend to dynamic environments to maintain `TRAJECT-Bench` in the long run.

## ACKNOWLEDGMENT

Pengfei He and Jiliang Tang are supported by the National Science Foundation (NSF) under grant numbers CNS2321416, IIS2212032, IIS2212144, IIS 2504089, DUE2234015, CNS2246050, and DRL2405483, the Michigan Department of Agriculture and Rural Development, US Dept of Commerce, Gates Foundation, Amazon Faculty Award, Meta, NVIDIA, Microsoft and SNAP. Yue Xing is supported by NSF DMS 2515194, Open Philanthropy, NVIDIA Academic Grant Program and Google Cloud Research Credit.

ETHICS STATEMENT

We acknowledge the ICLR Code of Ethics and ensure that no concerns regarding the Code of Ethics arise from our work. Our study does not involve human subjects, personal or sensitive data, or experiments that could cause harm. All data used are either synthetic or publicly available under appropriate licenses, and we have adhered to principles of fairness, transparency, and reproducibility throughout.

REPRODUCIBILITY STATEMENT

We provide data and codes in `https://anonymous.4open.science/r/ToolData-public-2565`, mentioned in section 4.1. We provide additional details in Appendix B.

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

# A   TRAJECT-BENCH DETAILS

We provide more details of the proposed TRAJECT-Bench.

## A.1   TOOL SET DETAILS AND EXAMPLES

**Overall statictics**. We provide statistics of the tool set in TRAJECT-Bench in Table 8.

Table 8: Statistics of tool set in TRAJECT-Bench

| domain | tool num |
|:---:|:---:|
| **Travel** | 131 |
| **Mapping** | 124 |
| **Finance** | 258 |
| **Weather** | 135 |
| **eCommerce** | 82 |
| **Mews_Media** | 118 |
| **Gaming** | 84 |
| **Email** | 70 |
| **Education** | 89 |
| **Music** | 137 |
| **Total** | 1228 |

**Tool examples**. We then provide some examples of tools in TRAJECT-Bench as follows.

```
{
  "tool name": "Airbnb listings: Prices and Availability by lat/lng",
  "tool description": "Retrieves average prices, availability percentages
      , and processed property counts within a 20,000-meter radius of a
      geographical point, filterable by bedrooms and guest capacity. This
       analytics endpoint leverages the parent tool's access to daily-
      updated Airbnb data, providing insights into property availability
      and pricing trends to help users make informed decisions about
      accommodations.",
  "required_parameters": [
    { "name": "year", "type": "NUMBER", "description": "the year", "
        default": 2024 },
    { "name": "lat", "type": "NUMBER", "description": "latitude", "
        default": 28.0852473 },
    { "name": "lng", "type": "NUMBER", "description": "longitude", "
        default": -16.7349705 },
    { "name": "range", "type": "NUMBER", "description": "range in meters
        from latitude and longitude point", "default": 500 },
    { "name": "month", "type": "STRING", "description": "the month", "
        default": "1" }
  ],
  "optional_parameters": [
    { "name": "bedrooms", "type": "NUMBER", "description": "number of
        bedrooms", "default": 1 },
    { "name": "maxguestcapacity", "type": "NUMBER", "description": "
        maximum guest capacity", "default": 2 }
  ],
  "API name": "Prices and Availability by lat/lng",
  "domain name": "Travel",
  "output_info": {
    "output_info_summary": "The tool provides a statistical summary of
        Airbnb listings within a specified geographical radius. A
        successful response returns aggregated data, including the total
        number of properties, median prices for all and for available
        properties, and the overall availability percentage. This
        information is intended to help users understand local
        accommodation market trends. The tool also returns a simple error
```

```
              message object in cases of failure, such as an authentication or
              subscription issue.",
       "output_format": "The output is a single, flat JSON object. The
              structure differs for successful and failed requests. A
              successful response contains keys like `totalProperties`, `
              medianPrice`, and `availability`. An error response typically
              contains a single `message` key with a string value. There is no
              indication of nesting, arrays, or pagination."
    },
    "connected tools": [
      {
        "tool name": "Airbnb listings: Listings by lat/lng",
        "connect params": [
          { "name": "lat", "description": "Can use the same latitude from
              the source tool's query" },
          { "name": "lng", "description": "Can use the same longitude from
              the source tool's query" },
          { "name": "range", "description": "Can use the same or adjusted
              range based on property density from source tool" }
        ]
      },
      {
        "tool name": "Priceline com Provider: Search hotels locations by
              geolocation",
        "connect params": [
          { "name": "latitude", "description": "Can use the same latitude
              from the source tool's query" },
          { "name": "longitude", "description": "Can use the same longitude
              from the source tool's query" }
        ]
      }
    ]
},
{
    "tool name": "Billboard: Hot 100",
    "tool description": "Provides detailed information about Billboard's
        Hot 100 chart, displaying the current rankings of the most
        popular songs in the United States. This endpoint delivers access
         to Billboard's comprehensive music chart system, which includes
        not only the Hot 100 but also other major charts like Billboard
        200, Billboard Global 200, and Artist 100. Billboard charts are
        the music industry's standard for measuring the popularity of
        artists, albums, and songs based on sales, radio airplay,
        streaming data, and other metrics, offering authoritative
        insights into current music trends and performance.",
    "required_parameters": [
      {
        "name": "date",
        "type": "DATE (YYYY-MM-DD)",
        "description": "Set the date for which you want to get a chart.",
        "default": "2020-03-18"
      }
    ],
    "optional_parameters": [],
    "API name": "Hot 100",
    "domain name": "Music",
    "output_info": {
      "output_info_summary": "The tool returns a ranked list of songs
          from the Billboard Hot 100 chart. For each song, it provides
          details such as the artist, title, current rank, last week's
          rank, peak position, and the number of weeks it has been on the
           chart. This data allows users to track the current popularity
          and historical performance of top songs in the United States.",
       "output_format": "The output is a string-serialized JSON array of
          objects. Each object corresponds to a song and its chart data.
```

```
            The 'artist' field contains HTML '<a>' tags linking to
            Billboard's website. The 'weeks_on_chart' field can be an empty
             string, likely for new entries. Some string values may contain
             HTML character entities (e.g., ''')."
    },
    "connected tools": [
      {
        "tool name": "Spotify: Search",
        "connect params": [
          {
            "name": "q",
            "description": "The song title and artist from Hot 100 can be
                used as search query"
          },
          {
            "name": "type",
            "description": "Can be set to 'tracks' to search for songs
                from the Hot 100 chart"
          }
        ]
      },
      {
        "tool name": "Shazam: search",
        "connect params": [
          {
            "name": "term",
            "description": "The song title and artist from Hot 100 can be
                used as search term"
          }
        ]
      },
      {
        "tool name": "Youtube Music API (Detailed): Search",
        "connect params": [
          {
            "name": "query",
            "description": "The song title and artist from Hot 100 can be
                used as search query"
          }
        ]
      }
    ]
  }
```

**Execution errors of tools**. As we mentioned in Section 3.1

## A.2 TRAJECTORY AND QUERY DETAILS

We provide more details of tool-use trajectories and queries in TRAJECT-Bench.

**Trajectory statistics**. The statistics of queries and trajectories in TRAJECT-Bench are shown in Table 9.

**Examples of task types for parallel queries**. We provide some examples of task types used in parallel query generations.

```
{
    "domain": "Travel",
    "task name": "Find Proper Accommodations",
    "task description": "Find proper hotels based on some options.  It
        requires searching across different platforms, comparing options
        based on real-time data, and retrieving specific, granular
        details like policies, recent reviews, or room availability that
        are crucial for making a booking decision.",
```

Table 9: Statistics of queries and trajectories in `TRAJECT-Bench`

| domain | simple parallel | hard parallel | sequential |
|---|---|---|---|
| **Travel** | 200 | 200 | 170 |
| **Mapping** | 200 | 200 | 200 |
| **Finance** | 200 | 200 | 135 |
| **Weather** | 200 | 200 | 200 |
| **eCommerce** | 200 | 200 | 200 |
| **Mews_Media** | 200 | 200 | 180 |
| **Gaming** | 200 | 200 | 185 |
| **Email** | 200 | 200 | 200 |
| **Education** | 200 | 200 | 200 |
| **Music** | 200 | 200 | 200 |
| **Total** | 2000 | 2000 | 1870 |

```
    "tool classes": [
        "Hotel",
        "Airbnb"
    ],
    "task examples": [
        {
            "query": "I'm planning a trip to London from Oct 10 to Oct 17
                in 2025. I need to find the best hotel deal. Can you
                search on both Booking.com and Hotels.com? I want a hotel
                 with at least a 4-star rating and I need to see its
                cancellation policy and some recent photos before I
                decide.",
            "tool list": [
                "Best Booking.com Hotel: find best booking.com
                    accommodation",
                "Hotels com Provider: Hotels Search",
                "Booking.com_v2: stays/properties/detail/policies",
                "Booking.com_v2: stays/properties/detail/photos"
            ]
        }
    ]
},
{
    "domain": "Gaming",
    "task name": "Tabletop & Strategy Game Assistance",
    "task description": "This task focuses on the non-video game tools,
        providing a user with rule-based assistance for a traditional
        tabletop or card game, such as calculating odds or determining
        the mathematically optimal move.",
    "tool classes": [
        "Card_Dice_Games",
        "Chess_Puzzles"
    ],
    "task examples": [
        {
            "query": "I'm playing a game of Blackjack and need some help.
                 The dealer is showing a 7. My hand is two 8s. According
                to basic strategy, what is the optimal move? Also, I
                think the deck is rich in high cards; the running count
                is +8 and there are about 4 decks left in the shoe. What
                is the True Count?",
            "tool list": [
                "BlackJack Basic Strategy: Pairs",
                "BlackJack Basic Strategy: True Count"
            ]
        }
    ]
},
```

```
{
    "domain": "Finance",
    "task name": "In-Depth Company Analysis",
    "task description": "This task involves creating a holistic financial
        overview of a specific company/stock by combining its reported
        fundamentals, current market data, and the consensus opinion of
        professional analysts.",
    "tool classes": [
        "Fundamental_Data",
        "Analyst_Ratings_Estimates"
    ],
    "task examples": [
        {
            "query": "Give me a complete financial workup on Apple (AAPL)
                . I want to see their latest quarterly income statement
                and balance sheet. Also, pull the current institutional
                ownership percentage and any recent insider transactions.
                 Finally, what is the consensus analyst price target for
                the stock?",
            "tool list": [
                "Financial Modeling Prep: Company Income Statement",
                "Financial Modeling Prep: Companies balance sheet
                    statement",
                "Mboum Finance: stock/institution-ownership",
                "EOD Historical Data: Insider Transactions API",
                "YFINANCE - Stock Market Data: Analyst Price Target"
            ]
        }
    ]
}
```

**Examples of task sequences for sequential queries**. We also provide some examples of task sequences used to generate exact sequential queries. It is clear that we define a coherence chain for each task and specify connections (parameter used for subsequent tools) among tools. Note that these sequences are general and do not contain any exact information such as detailed parameter values, so they can serve as templates to create exact trajectories and queries fitting different realistic scenarios.

```
{
    "domain": "Music",
    "name": "Radio Genre to Artist Profile Discovery",
    "description": "This sequence starts by discovering radio stations in
        a specific country to identify a music genre, then finds an
        artist in that genre on Spotify, and concludes by retrieving that
        artist's detailed profile.",
    "tool sequence": [
    {
    "tool name": "50K Radio Stations: Get Countries",
    "use description": "Retrieves a list of countries with radio stations
        to select a country ID.",
    "param for next tool": "country_id"
    },
    {
    "tool name": "50K Radio Stations: Get Channels",
    "use description": "Finds radio channels in the selected country to
        identify a prominent genre.",
    "param for next tool": "q"
    },
    {
    "tool name": "Spotify: Search",
    "use description": "Searches for artists on Spotify using the genre
        as a query to get an artist's ID.",
    "param for next tool": "artist_id"
    },
```

```
    {
    "tool name": "Spotify _v2: Artist Details (Single)",
    "use description": "Fetches the complete profile for the artist from
        Spotify using their ID.",
    "param for next tool": ""
    }
    ],
    "tool number": 4
},
{

    "domain": "Education",
    "name": "DEI Resource -- Speaker Analysis -- Vocabulary Deep Dive --
        Further Study",
    "description": "This sequence models a student's workflow: finding a
        DEI resource, identifying a speaker, finding their TED talks, and
         then performing a deep vocabulary analysis on a key term before
        exploring related learning resources.",
    "tool sequence": [
    {
    "tool name": "DEI: GetLearningResources",
    "use description": "A user gets a list of DEI learning resources and
        selects one, which provides a 'title'.",
    "param for next tool": "title"
    },
    {
    "tool name": "DEI: GetPeople",
    "use description": "The 'title' is used to find an influential person
         related to that topic, providing their 'name'.",
    "param for next tool": "name"
    },
    {
    "tool name": "TED Talks API: getTalks",
    "use description": "The person's 'name' is used as the 'speaker' to
        find their TED talks, which returns a 'talk_title'.",
    "param for next tool": "talk_title"
    },
    {
    "tool name": "Urban Dictionary: Define a Word / Search",
    "use description": "A key term from the 'talk_title' is used as the '
        term' to look up its modern, colloquial, or slang meaning.",
    "param for next tool": "term"
    },
    {
    "tool name": "Dictionary: wordSearchTurkish",
    "use description": "The same 'term' is used as a 'query' to find its
        Turkish translation, providing an 'anlam' (meaning).",
    "param for next tool": "anlam"
    },
    {
    "tool name": "Urban Dictionary: Define a Word / Search",
    "use description": "The Turkish 'anlam' (meaning) is used as a 'term'
         in a reverse-lookup to see its English slang definitions.",
    "param for next tool": "topic"
    },
    {
    "tool name": "DEI: GetLearningResources",
    "use description": "Finally, the original 'topic' from the talk is
        used to find other, related learning resources.",
    "param for next tool": ""
    }
    ],
    "tool number": 7
}
```

**Examples of queries and trajectories**. We then provide examples of trajectories and queries in
`TRAJECT-Bench`.

*Parallel.*

```
Example 1:  Travel domain
Tool Trajectory: [
      {
        "tool name": "Priceline com Provider: Hotel reviews",
        "tool description": "Returns a list of reviews for travel
            services such as hotels, cars, and flights. This endpoint is
            part of the Priceline.com API, which enables users to search
            and book travel accommodations and rentals. It leverages a
            database of user-generated reviews to provide insights,
            helping travelers make informed decisions about their
            bookings.",
        "required parameters": [
          {
            "name": "hotel_id",
            "value": "700022612"
          }
        ],
        "optional parameters": [
          {
            "name": "languages",
            "value": "en,fr"
          },
          {
            "name": "limit",
            "value": 100
          },
          {
            "name": "only_verified_guests",
            "value": true
          }
        ],
        "executed_output": "{'getHotelReviews': {'error': {'status': '
            Hotel.Reviews: No reviews found as Offset value exceeds the
            number of reviews.', 'status_code': '1.822.8', 'time':
            '0.0715'}}}"
      },
      {
        "tool name": "Booking.com_v2: languages",
        "tool description": "Retrieves available language options for
            hotel information and booking interfaces. This endpoint
            leverages Booking.com's unofficial API to access real-time
            data from hotels worldwide, allowing users to specify their
            preferred language when querying room availability, pricing,
            facilities, and policies as part of comprehensive travel
            planning.",
        "required parameters": [],
        "optional parameters": [],
        "executed_output": "{'data': [{'__ref': 'Language:{\"code\":\"en-
            gb\"}'}, {'__ref': 'Language:{\"code\":\"en-us\"}'}, {'__ref
            ': 'Language:{\"code\":\"de\"}'}, {'__ref': 'Language:{\"code
            \":\"nl\"}'}, {'__ref': 'Language:{\"code\":\"fr\"}'}, {'
            __ref': 'Language:{\"code\":\"es\"}'}, {'__ref': 'Language
            :{\"code\":\"es-ar\"}'}, {'__ref': 'Language:{\"code\":\"es-
            mx\"}'}, {'__ref': 'Language:{\"code\":\"ca\"}'}, {'__ref': '
            Language:{\"code\":\"it\"}'}, {'__ref': 'Language:{\"code
            \":\"pt-pt\"}'}, {'__ref': 'Language:{\"code\":\"pt-br\"}'},
            {'__ref': 'Language:{\"code\":\"no\"}'}, {'__ref': 'Language
            :{\"code\":\"fi\"}'}, {'__ref': 'Language:{\"code\":\"sv
            \"}'}, {'__ref': 'Language:{\"code\":\"da\"}'}, {'__ref': '
            Language:{\"code\":\"cs\"}'}, {'__ref': 'Language:{\"code
```

\":\"hu\"}'}, {'__ref': 'Language:{\"code\":\"ro\"}'}, {'__ref': 'Language:{\"code\":\"ja\"}'}, {'__ref': 'Language:{\"code\":\"zh-cn\"}'}, {'__ref': 'Language:{\"code\":\"zh-tw\"}'}, {'__ref': 'Language:{\"code\":\"pl\"}'}, {'__ref': 'Language:{\"code\":\"el\"}'}, {'__ref': 'Language:{\"code\":\"ru\"}'}, {'__ref': 'Language:{\"code\":\"tr\"}'}, {'__ref': 'Language:{\"code\":\"bg\"}'}, {'__ref': 'Language:{\"code\":\"ar\"}'}, {'__ref': 'Language:{\"code\":\"ko\"}'}, {'__ref': 'Language:{\"code\":\"he\"}'}, {'__ref': 'Language:{\"code\":\"lv\"}'}, {'__ref': 'Language:{\"code\":\"uk\"}'}, {'__ref': 'Language:{\"code\":\"hi\"}'}, {'__ref': 'Language:{\"code\":\"id\"}'}, {'__ref': 'Language:{\"code\":\"ms\"}'}, {'__ref': 'Language:{\"code\":\"th\"}'}, {'__ref': 'Language:{\"code\":\"et\"}'}, {'__ref': 'Language:{\"code\":\"hr\"}'}, {'__ref': 'Language:{\"code\":\"lt\"}'}, {'__ref': 'Language:{\"code\":\"sk\"}'}, {'__ref': 'Language:{\"code\":\"sr\"}'}, {'__ref': 'Language:{\"code\":\"sl\"}'}, {'__ref': 'Language:{\"code\":\"vi\"}'}, {'__ref': 'Language:{\"code\":\"tl\"}'}, {'__ref': 'Language:{\"code\":\"is\"}'}], 'message': 'Successful', 'status': True}"
},
{
  "tool name": "Cities Cost of Living: Get Cities List",
  "tool description": "Retrieves a comprehensive list of all available cities in the database. This endpoint taps into a global dataset covering over 650 cities worldwide, providing access to the complete catalog of locations for which detailed living expense information is available. The parent service offers extensive cost-of-living data that helps users compare and understand financial requirements across different urban centers globally.",
  "required parameters": [],
  "optional parameters": [],
  "executed_output": "{'cities': [{'country': 'Turkey', 'name': 'Kocaeli'}, {'country': 'Czech Republic', 'name': 'Ostrava'}, {'country': 'United States', 'name': 'Santa Clara'}, {'country': 'United States', 'name': 'Iowa City'}, {'country': 'United Kingdom', 'name': 'Bournemouth'}, {'country': 'Montenegro', 'name': 'Podgorica'}, {'country': 'United States', 'name': 'Toledo'}, {'country': 'United Kingdom', 'name': 'Milton Keynes'}, {'country': 'India', 'name': 'Mangalore'}, {'country': 'Malaysia', 'name': 'Johor Bahru'}, {'country': 'Canada', 'name': \"St. John's\"}, {'country': 'Pakistan', 'name': 'Islamabad'}, {'country': 'Norway', 'name': 'Trondheim'}, {'country': 'India', 'name': 'Nagpur'}, {'country': 'Czech Republic', 'name': 'Olomouc'}, {'country': 'Netherlands', 'name': 'Best'}, {'country': 'Ukraine', 'name': 'Zaporizhzhya'}, {'country': 'India', 'name': 'Mysore'}, {'country': 'China', 'name': 'Beijing'}, {'country': 'North Macedonia', 'name': 'Skopje'}, {'country': 'Philippines', 'name': 'Manila'}, {'country': 'Netherlands', 'name': 'Rotterdam'}, {'country': 'Taiwan (China)', 'name': 'Taipei'}, {'country': 'Brazil', 'name': 'Goiania'}, {'country': 'Taiwan (China)', 'name': 'New Taipei City'}, {'country': 'Austria', 'name': 'Salzburg'}, {'country': 'Germany', 'name': 'Frankfurt'}, {'country': 'Turkey', 'name': 'Diyarbakir'}, {'country': 'Canada', 'name': 'North Vancouver'}, {'country': 'India', 'name': 'Ludhiana'}, {'country': 'Israel', 'name': 'Jerusalem'}, {'country': 'Afghanistan', 'name': 'Kabul'}, {'country': 'Romania', 'name': 'Iasi'}, {'country': 'United Kingdom', 'name': 'Bristol'}, {'country': 'South Africa', 'name': 'Port Elizabeth'}, {'country': 'Ecuador', 'name': 'Guayaquil'}, {'country': 'France', 'name': 'Bordeaux'}, {'country': 'Georgia', 'name': 'Batumi'}, {'country': 'Botswana

```
            ', 'name': 'Gaborone'}, {'country': 'Brazil', 'name': 'Natal
            '}, {'country': 'Trinidad And Tobago', 'name': 'Port of Spain
            '}, {'country': 'Belgium', 'name': 'Antwerp'}, {'country': '
            Austria', 'name': 'Vienna'}, {'country': 'France', 'name': "
      }
    ],
    "trajectory_type": "parallel",
    "tool count": 3,
    "final_answer": "# Travel Logistics Information\n\n## 1. Hotel
        Reviews from Priceline.com\n\nThe Priceline.com Provider tool
        returned an error when attempting to retrieve hotel reviews. The
        specific error message was:\n\n\"Hotel.Reviews: No reviews found
         as Offset value exceeds the number of reviews.\"\n\nThis suggests
         that either the hotel ID provided (700022612) doesn't have any
        reviews, or there was an issue with the parameters used in the
        request. To get valid hotel reviews, you would need to provide a
        different hotel ID or adjust the offset parameter.\n\n## 2.
        Languages Supported by Booking.com\n\nBooking.com supports 45
        languages across various regions. The complete list includes:\n\n
        - English (UK): en-gb\n- English (US): en-us\n- German: de\n-
        Dutch: nl\n- French: fr\n- Spanish: es\n- Spanish (Argentina): es
        -ar\n- Spanish (Mexico): es-mx\n- Catalan: ca\n- Italian: it\n-
        Portuguese (Portugal): pt-pt\n- Portuguese (Brazil): pt-br\n-
        Norwegian: no\n- Finnish: fi\n- Swedish: sv\n- Danish: da\n-
        Czech: cs\n- Hungarian: hu\n- Romanian: ro\n- Japanese: ja\n-
        Chinese (Simplified): zh-cn\n- Chinese (Traditional): zh-tw\n-
        Polish: pl\n- Greek: el\n- Russian: ru\n- Turkish: tr\n-
        Bulgarian: bg\n- Arabic: ar\n- Korean: ko\n- Hebrew: he\n-
        Latvian: lv\n- Ukrainian: uk\n- Hindi: hi\n- Indonesian: id\n-
        Malay: ms\n- Thai: th\n- Estonian: et\n- Croatian: hr\n-
        Lithuanian: lt\n- Slovak: sk\n- Serbian: sr\n- Slovenian: sl\n-
        Vietnamese: vi\n- Tagalog: tl\n- Icelandic: is\n\n## 3. Cities
        for Cost-of-Living Comparisons\n\nThe Cities Cost of Living tool
        returned a partial list of cities available for cost-of-living
        comparisons. The output was truncated, showing only the first 43
        cities out of what appears to be a much larger database (
        reportedly covering over 650 cities worldwide). \n\nSome of the
        cities included in the partial list:\n- Kocaeli, Turkey\n-
        Ostrava, Czech Republic\n- Santa Clara, United States\n- Iowa
        City, United States\n- Bournemouth, United Kingdom\n- Beijing,
        China\n- Manila, Philippines\n- Frankfurt, Germany\n- Vienna,
        Austria\n\nPlease note that this is only a small subset of the
        complete city list, as the tool output was cut off after 43
        cities.\n\n# Summary\n\nI've gathered information on languages
        supported by Booking.com (45 languages) and a partial list of
        cities available for cost-of-living comparisons. Unfortunately,
        the hotel reviews request returned an error, so no review data is
         available with the provided parameters. To complete your travel
        logistics preparation, you may need to request hotel reviews with
         different parameters and obtain the complete city list for cost-
        of-living comparisons."
}
```

Simple query: I'm preparing travel logistics. Can you grab: Hotel reviews
     from Priceline.com; which languages Booking.com supports; and the
    city list for cost of living comparisons?

Hard query: I'm pulling together travel logistics for a fall trip. Could
    you first scan recent guest feedback from a major U.S. hotel
    aggregator and summarize consistent pros/cons for central options?
    Next, check which languages the leading global booking portal
    actually supports across its site and app (not just auto-translate),
    and note any gaps. Finally, put together a short list of major cities
     we can use for side-by-side cost-of-living comparisons.

**Example 2:  eCommerce domain**
Tool Trajectory: [

```
{
  "tool name": "Wayfair: auto-complete",
  "tool description": "Retrieves auto suggestions based on a
      provided term or phrase, helping users quickly find relevant
      content. This API provides comprehensive search functionality
       for Wayfair's extensive catalog, allowing you to query for
      information about categories, products, and other offerings
      just as you would on wayfair.com. The service streamlines the
       discovery process by suggesting relevant terms as users type
      , making it easier to navigate through Wayfair's extensive
      home goods and furniture inventory.",
  "required parameters": [
    {
      "name": "query",
      "value": "red office chair"
    }
  ],
  "optional parameters": [],
  "executed_output": "{'summary': {'request_id': '', '
      transaction_id': 'u4aD+FSTT4S/epB1PGO3bw==', 'page_type': '
      General', 'response_hash': '231
      be424e42fe70df57916c01134581d1986d7a4', '
      response_matches_prior_hash': False, 'cache_seconds': 1800, '
      cache_always_check_server': False, 'spv_custom_vars': '
      LoginStatusFlag=0'}, 'response': [{'schema_id': '
      WFSearchSuggestion', 'value': 'red office chair', 'type': '
      keyword', 'first_in_section': False, 'is_reform': False, '
      keyword': False}, {'schema_id': 'WFSearchSuggestion', 'value
      ': 'office chair red', 'type': 'keyword', 'first_in_section':
       False, 'is_reform': False, 'keyword': False}, {'schema_id':
      'WFSearchSuggestion', 'value': 'red office chairs with gold
      frames', 'type': 'keyword', 'first_in_section': False, '
      is_reform': False, 'keyword': False}]}"
},
{
  "tool name": "Asos: v2/auto-complete",
  "tool description": "Gets autocomplete suggestions based on a
      partial product name input, helping users quickly find
      specific items they're looking for. This endpoint leverages
      the Asos API's comprehensive product database to deliver
      relevant search completions, functioning just like the
      autocomplete feature on the official Asos website. The API
      provides access to the same extensive catalog of categories,
      products, and related information that powers the Asos
      shopping platform.",
  "required parameters": [
    {
      "name": "q",
      "value": "bikini top"
    }
  ],
  "optional parameters": [
    {
      "name": "store",
      "value": "US"
    },
    {
      "name": "country",
      "value": "US"
    },
    {
      "name": "currency",
      "value": "USD"
    },
    {
```

```
              "name": "lang",
              "value": "en-US"
          }
      ],
      "executed_output": "{'suggestionGroups': [{'indexName': '
          searchterms', 'indexTitle': 'searchterms', 'suggestions': [{'
          searchTerm': 'bikini top', 'numberOfResults': 2460}, {'
          searchTerm': 'underwire bikini top', 'numberOfResults': 213},
           {'searchTerm': 'bandeau bikini top', 'numberOfResults':
          225}, {'searchTerm': 'black bikini top', 'numberOfResults':
          414}, {'searchTerm': 'fuller bust bikini top', '
          numberOfResults': 95}, {'searchTerm': 'white bikini top', '
          numberOfResults': 303}, {'searchTerm': 'triangle bikini top',
           'numberOfResults': 519}]}]}"
    },
    {
      "tool name": "Asos: categories/list",
      "tool description": "Lists all available product categories from
          Asos, providing a structured overview of the shopping
          taxonomy. This endpoint taps into the comprehensive Asos API
          system that mirrors the official website's data architecture,
           allowing users to efficiently navigate through the complete
          category hierarchy before diving into specific product
          searches or filtering options.",
      "required parameters": [],
      "optional parameters": [
          {
              "name": "lang",
              "value": "en-US"
          },
          {
              "name": "country",
              "value": "US"
          }
      ],
      "executed_output": "{'navigation': [{'id': '7276d7f9-b810-4743-8
          c11-eccb260bbecd', 'alias': 'MW', 'type': 'link', '
          channelExclusions': [], 'webLargePriority': 0, 'content': {'
          title': 'Men', 'subTitle': None, 'webLargeImageUrl': None, '
          mobileImageUrl': None}, 'display': None, 'style': {'
          webLargeStyleType': 'dark', 'mobileStyleType': 'dark'}, 'link
          ': {'linkType': 'internal', 'brandSectionAlias': None, '
          categoryId': None, 'webUrl': 'https://www.asos.com/us/men/',
          'appUrl': None}, 'children': [{'id': 'ae28af2b-e3ca-4f2f-a559
          -9a976a0812d4', 'alias': None, 'type': 'container', '
          channelExclusions': [], 'webLargePriority': 0, 'content': {'
          title': 'Home', 'subTitle': None, 'webLargeImageUrl': None, '
          mobileImageUrl': None}, 'display': {'webLargeTemplateId': 0,
          'webLargeTemplateName': '', 'webLargeColumnSpan': 1, '
          mobileTemplateId': 7, 'mobileTemplateName': 'homepageLinks',
          'mobileDisplayLayout': 'list'}, 'style': {'webLargeStyleType
          ': 'dark', 'mobileStyleType': 'noTitle'}, 'link': None, '
          children': [{'id': '47989db5-0597-43e6-9bee-1348c35590b0', '
          alias': None, 'type': 'link', 'channelExclusions': ['apps', '
          webLarge'], 'webLargePriority': 0, 'content': {'title': 'Home
          ', 'subTitle': None, 'webLargeImageUrl': 'https://images.asos
          -media.com/navigation/mw_homebuttonnew_1826147', '
          mobileImageUrl': 'https://images.asos-media.com/navigation/
          mw_homebuttonnew_1826147'}, 'display': None, 'style': {'
          webLargeStyleType': 'dark', 'mobileStyleType': 'dark'}, 'link
          ': {'linkType': 'internal', 'brandSectionAlias': None, '
          categoryId': None, 'webUrl': 'https://www.asos.com/us/men/',
          'appUrl': None}, 'children': []}]}, {'id': 'da3b6d82-c0e8-423
          f-a6ed-f23e930ce951', 'alias': None, 'type': 'container', '
          channelExclusions': ['webLarge'], 'webLargePriority': 0, '
```

```
            content': {'title': 'App and Mobile Top Level - Carousel', '
            subTitle': None, 'webLargeImageUrl': None, 'mobileImageUrl':
            None}, 'display': {'webLargeTemplateId': 0, '
            webLargeTemplateName': '', 'webLargeColumnSpan': 1, '
            mobileTemplateId': 1, 'mobileTemplateName': 'imageBackground
            ', 'mobileDisplayLayout': 'list'"
        }
    ],
    "trajectory_type": "parallel",
    "tool count": 3,
    "final_answer": "# Retailer Comparison: Wayfair and Asos\n\n##
        Wayfair Search Results for 'red office chair'\nI used Wayfair's
        auto-complete tool to search for 'red office chair' and found
        these suggested search terms:\n- red office chair\n- office chair
         red\n- red office chairs with gold frames\n\nThe tool only
        provided search suggestions and did not return actual product
        listings, prices, or availability information.\n\n## Asos Search
        Results for 'bikini top'\nI searched Asos for 'bikini top' using
        the US store, US country setting, USD currency, and English
        language. The auto-complete tool returned these suggestions with
        their respective product counts:\n- bikini top (2,460 products)\n
        - underwire bikini top (213 products)\n- bandeau bikini top (225
        products)\n- black bikini top (414 products)\n- fuller bust
        bikini top (95 products)\n- white bikini top (303 products)\n-
        triangle bikini top (519 products)\n\nThis indicates Asos has a
        substantial selection of bikini tops with various styles
        available.\n\n## Asos Categories\nI attempted to retrieve the
        full category list from Asos for the US market, but the tool
        returned only partial information. The output shows the beginning
         of the men's category structure but was cut off. The data
        includes:\n- Main category: Men\n- Some navigation structure
        information\n- Partial subcategory information\n\nThe tool did
        not return the complete category list, particularly missing women
        's categories which would be most relevant for bikini tops.\n\n##
         Summary\nThis comparison provides a limited snapshot of both
        retailers. For Wayfair, we only have search suggestions for red
        office chairs without actual product information. For Asos, we
        have quantitative data showing they offer over 2,400 bikini top
        products with various style options, but the category listing was
         incomplete. To make a comprehensive comparison between these
        retailers, additional information about actual products, pricing,
         shipping options, and return policies would be needed."
}
Simple query: I'm comparing retailers and want a clear snapshot. Please
    get Wayfair auto-complete suggestions for 'red office chair', and get
     Asos auto-complete suggestions for 'bikini top' with store US,
    country US, currency USD, lang en-US, and list Asos categories with
    country US, lang en-US.
Hard query:We're moving next month and I'm trying to stretch a tight
    budget without buying junk. Here are the references I've jotted down:
     On Wayfair, I'm thinking along the lines of 'red office chair' show
    what's trending. On Asos since I'm shopping from the US and paying in
     dollars, I'm exploring 'bikini top' to round out accessories. I want
     to see how Asos organizes things for a US selection in English so I
    don't miss a section. If something is clearly better value, flag it\
    u2014otherwise show me the top few comparable picks.
```

*Sequential.*

```
Query: "I'm looking for Ed Sheeran's \"Shape of You\" on SoundCloud. Can
    you find the track, then get detailed metadata for it using the
    official SoundCloud URL, and finally download the timed lyrics by
    matching it with the third candidate on Spotify? I want the lyrics
    for track 301161123 since it has over 12,000 comments and is one of
    his most popular songs from 2017.",
```

```
Tool trajectory: [
    {
      "tool name": "Miza: Song search",
      "tool description": "Searches for a song on SoundCloud using the
          'scsearch:' prefix to get its direct URL.",
      "required parameters": [
        {
          "name": "search",
          "value": "scsearch:Shape of You Ed Sheeran"
        }
      ],
      "optional parameters": [],
      "execution_status": "success",
      "executed_output": "[{'name': 'Shape of You', 'url': 'https://
          soundcloud.com/edsheeran/shape-of-you', 'duration': 233.759,
          'icon': 'https://i1.sndcdn.com/artworks-jnr3tXcz4dKQ-0-
          original.jpg'}]",
      "API name": "Song search",
      "domain name": "Music",
      "parent tool name": "Miza",
      "sequence_step": {
        "step_number": 1,
        "tool_name": "Miza: Song search",
        "description": "Searches for a song on SoundCloud using the '
            scsearch:' prefix to get its direct URL.",
        "param_for_next_tool": "track",
        "original_description": "{'tool name': 'Miza: Song search', '
            use description': \"Searches for a song on SoundCloud using
             the 'scsearch:' prefix to get its direct URL.\", 'param
            for next tool': 'track'}"
      },
      "original_description": "{'tool name': 'Miza: Song search', 'use
          description': \"Searches for a song on SoundCloud using the '
          scsearch:' prefix to get its direct URL.\", 'param for next
          tool': 'track'}"
    },
    {
      "tool name": "SoundCloud Scraper: Get Track Metadata (1-3 Quotas)
          ",
      "tool description": "Retrieves the full metadata for the
          SoundCloud track using its URL.",
      "required parameters": [
        {
          "name": "track",
          "value": "https://soundcloud.com/edsheeran/shape-of-you"
        }
      ],
      "optional parameters": [],
      "execution_status": "success",
      "executed_output": "{'status': True, 'errorId': 'Success', 'audio
          ': [{'quality': 'sq', 'url': 'https://scd.dlod.link/', '
          durationMs': 233744, 'durationText': '03:53', 'mimeType': '
          audio/mpeg', 'extension': 'mp3'}, {'quality': 'sq', 'url': '
          https://scd.dlod.link/', 'durationMs': 233719, 'durationText
          ': '03:53', 'mimeType': 'audio/ogg; codecs=\"opus\"', '
          extension': 'opus'}], 'type': 'track', 'id': 301161123, '
          permalink': 'https://soundcloud.com/edsheeran/shape-of-you',
          'createdAt': '2017-01-06T04:05:41Z', 'lastModified':
          '2025-09-05T06:03:33Z', 'title': 'Shape of You', 'caption':
          None, 'description': None, 'goPlus': False, 'durationMs':
          233744, 'durationText': '03:53', 'artworkUrl': 'https://i1.
          sndcdn.com/artworks-cK0SqfJHnCqU-0-original.jpg', '
          releaseDate': '2017-01-06T00:00:00Z', 'genre': 'Pop', '
          labelName': 'Atlantic Records UK', 'license': 'all-rights-
```

```
        reserved', 'purchaseTitle': None, 'purchaseUrl': None, '
        commentable': True, 'commentCount': 12127, 'likeCount",
    "API name": "Get Track Metadata (1-3 Quotas)",
    "domain name": "Music",
    "parent tool name": "SoundCloud Scraper",
    "sequence_step": {
      "step_number": 2,
      "tool_name": "SoundCloud Scraper: Get Track Metadata (1-3
          Quotas)",
      "description": "Retrieves the full metadata for the SoundCloud
          track using its URL.",
      "param_for_next_tool": "track",
      "original_description": "{'tool name': 'SoundCloud Scraper: Get
           Track Metadata (1-3 Quotas)', 'use description': '
          Retrieves the full metadata for the SoundCloud track using
          its URL.', 'param for next tool': 'track'}"
    },
    "original_description": "{'tool name': 'SoundCloud Scraper: Get
        Track Metadata (1-3 Quotas)', 'use description': 'Retrieves
        the full metadata for the SoundCloud track using its URL.', '
        param for next tool': 'track'}",
    "adapt_parameter": "track",
    "adapt_constraint": "track URL for a popular Ed Sheeran song",
    "adapt_reason": "Selected the URL for 'Shape of You' by Ed
        Sheeran as it represents the track identifier in the
        SoundCloud ecosystem. This URL uniquely identifies the track
        and is the most appropriate parameter for the SoundCloud
        Scraper's 'track' parameter."
  },
  {
    "tool name": "SoundCloud Scraper: Download Lyrics on Spotify (2
        Quotas)",
    "tool description": "Uses the SoundCloud track URL to find a
        match on Spotify and download its timed lyrics.",
    "required parameters": [
      {
        "name": "track",
        "value": "301161123"
      }
    ],
    "optional parameters": [
      {
        "name": "candidate",
        "value": 3
      }
    ],
    "execution_status": "success",
    "executed_output": "[00:09.77]The club isn't the best place to
        find a lover\n[00:12.16]So the bar is where I go, mm-mm\n
        [00:14.94]Me and my friends at the table doin' shots\n
        [00:17.16]Drinkin' fast and then we talk slow, mm-mm\n
        [00:19.41]And you come over and start up a conversation with
        just me\n[00:22.36]And trust me, I'll give it a chance now,
        mm-mm\n[00:24.68]Take my hand, stop, put Van the Man on the
        jukebox\n[00:27.36]And then we start to dance, and now I'm
        singin' like\n[00:29.95]Girl, you know I want your love\n
        [00:32.43]Your love was handmade for somebody like me\n
        [00:35.55]Come on now, follow my lead\n[00:37.40]I may be
        crazy, don't mind me\n[00:39.58]Say, \"Boy, let's not talk
        too much\"\n[00:42.44]Grab on my waist and put that body on
        me\n[00:45.59]Come on now, follow my lead\n[00:47.13]Come,
        come on now, follow my lead, mm-mm\n[00:50.88]I'm in love
        with the shape of you\n[00:53.21]We push and pull like a
        magnet do\n[00:55.76]Although my heart is fallin' too\n
        [00:58.35]I'm in love with your body\n[01:00.66]And last
```

```
              night, you were in my room\n[01:03.12]And now my bedsheets
              smell like you\n[01:05.35]Every day, discoverin' somethin'
              brand-new\n[01:08.13]Well, I'm in love with your body\n
              [01:10.02]Oh, I, oh, I, oh, I, oh, I\n[01:13.34]Well, I'm in
              love with your body\n[01:14.91]Oh, I, oh, I, oh, I, oh, I\n
              [01:18.20]Well, I'm in love with your body\n[01:19.92]Oh, I,
              oh, I, oh, I, oh, I\n[01:23.22]Well, I'm in love with your
              body\n[01:25.33]Every day, discoverin' somethin' brand-new\n
              [01:28.39]I'm in love with the shape ",
        "API name": "Download Lyrics on Spotify (2 Quotas)",
        "domain name": "Music",
        "parent tool name": "SoundCloud Scraper",
        "sequence_step": {
          "step_number": 3,
          "tool_name": "SoundCloud Scraper: Download Lyrics on Spotify (2
              Quotas)",
          "description": "Uses the SoundCloud track URL to find a match
              on Spotify and download its timed lyrics.",
          "param_for_next_tool": "",
          "original_description": "{'tool name': 'SoundCloud Scraper:
              Download Lyrics on Spotify (2 Quotas)', 'use description':
              'Uses the SoundCloud track URL to find a match on Spotify
              and download its timed lyrics.', 'param for next tool':
              ''}"
        },
        "original_description": "{'tool name': 'SoundCloud Scraper:
            Download Lyrics on Spotify (2 Quotas)', 'use description': '
            Uses the SoundCloud track URL to find a match on Spotify and
            download its timed lyrics.', 'param for next tool': ''}",
        "adapt_parameter": "track",
        "adapt_constraint": "track ID for a popular song with high
            engagement",
        "adapt_reason": "Selected 'Shape of You' by Ed Sheeran (ID:
            301161123) because it has significant engagement (12127+
            comments) and is a well-known track from 2017. The track ID
            is the unique identifier needed for the SoundCloud Scraper."
      }
    ],
    "domain": "Music",
    "final_answer": "I successfully found Ed Sheeran's 'Shape of You' on
        SoundCloud and retrieved the detailed metadata and timed lyrics
        as requested. Here's what I found:\n\n1. SoundCloud Search
        Results:\n   - Successfully located 'Shape of You' by Ed Sheeran\
        n   - URL: https://soundcloud.com/edsheeran/shape-of-you\n   -
        Duration: 233.759 seconds (approximately 3:53)\n2. Track
        Metadata from SoundCloud:\n   - Confirmed this is track ID
        301161123 as you mentioned\n   - The track has two audio quality
        options available (mp3 and opus formats)\n   - Duration matches
        the search result at approximately 3:53\n\n3. Timed Lyrics:\n   -
         Successfully retrieved the synchronized lyrics from Spotify (
        third candidate match as requested)\n   - The lyrics are time-
        stamped, starting at [00:09.77] with \"The club isn't the best
        place to find a lover\"\n   - The lyrics continue throughout the
        song with precise timestamps\n\nThe track you specified
        (301161123) was indeed found and appears to be the official
        version from Ed Sheeran's SoundCloud account, matching your
        description as one of his most popular songs from 2017."
```

## A.3 ADDITIONAL DISCUSSIONS

While we mainly discuss using TRAJECT-Bench as an evaluation and diagnosis framework for agentic tool usage, TRAJECT-Bench can go beyond that. The high-quality trajectories and labels can support multiple training and improvement paradigms.

a) Pre-training or post-training to enhance tool-use capabilities. Our dataset provides real-world task-driven queries and solution trajectories, so it is possible to be used for supervised training. The trajectory-aware metrics can also serve as rewards to enhance the model's capability to plan long tool sequences and avoid tool hallucinations.

b) In-context learning (ICL) demonstrations. The benchmark provides clean, labeled tool trajectories that can be placed in the context window to improve planning, tool selection and other capabilities.

c) Improving reasoning ability (long-chain + implicit reasoning). As shown in Figure 2 and Table 2, current models and agents still struggle with long tool chains and implicit user queries. Therefore, researchers can generate intermediate reasoning steps and tool calls built upon our data for training purposes.

We believe exploring these directions a promising future direction and requires significant efforts–training not only relies on data, but also relies on algorithms and many factors. We will keep investigating.

## B  ADDITIONAL EVALUATION DETAILS

In this section, we provide more details about the evaluation.

**Evaluation on individual LLMs**. For this evaluation, we provide both the test query and the available tools as context to the model and prompt the model to predict a trajectory. Specifically, we enforce the model to given answer in JSON format. The prompt is as follows.

```
Direct Prompt
Given the tool list:
<tools>

Please slove the query with help of these tools if necessary:
<query>

Please selct proper tools and, provide me proper parameters to call it.
    If you need outputs from previous tools, use a placeholder '<results
    from tool xxx>' .Please respond in the following json format:
'''json
{
\"tool list\": [
{\"tool name\":[tool name],
\"tool description\":[tool description],
\"required_parameters\":[{\"name\": xxx,\"value\": xxx}, {\"name\": xxx
    ,\"value\": xxx},...],
\"optional_parameters\":[{\"name\": xxx,\"value\": xxx},...],
\"API name\":[API name],
\"domain name\":[domain name],
\"parent tool name\":[parent tool name]},
...
]
}
'''
```

Despite the direct prompting, we also include CoT prompting.

```
CoT prompt
You are a problem solver.
Goal: given a user query and a pool of tools, produce the MINIMAL,
    CORRECT tool-call trajectory that satisfies the query.
Please think step by step to:
1) extract intent, entities, constraints, required outputs,
2) shortlist feasible tools (match inputs\u2192outputs),
3) design a trajectory (sequential or parallel),
4) compute arguments with correct types/formats/units,
5) check dependencies and stop criteria.
```

```
Here are some examples:
Query: Schedule dinner with Alice at 19:00\u201321:00, 2025-8-19 in San
    Jose at a vegan place; invite alice@example.com.
Tools: [\n  {\n    \"tool name\": \"Calendar: create_calendar_event\",\n
       \"tool description\": \"Create an event; emails participants.\",\n
        \"required_parameters\": [\n     {\n      \"name\": \"title\",\n
    \"type\": \"string\",\n     \"default\": \"set up a meeting\",\n      \"
    description\": \"The title of the event\",\n     },\n     {\n     \"name
    \": \"start_time\",\n     \"type\": \"string\",\n     \"default\":
    \"2025-8-19 19:00\",\n     \"description\": \"The start time of the
    event\",\n      },\n     {\n     \"name\": \"end_time\",\n     \"type\":
    \"string\",\n     \"default\": \"2025-8-19 21:00\",\n     \"description
    \": \"The end time of the event\",\n      },\n     {\n     \"name\": \"
    participants\",\n     \"type\": \"list<email>\",\n     \"default\": [\"
    alice@example.com\"],\n     \"description\": \"The participants of the
     event\",\n      }\n     ],\n     \"optional_parameters\": [\n     {\n
    \"name\": \"location\",\n     \"type\": \"str\",\n     \"default\": \"
    San Jose\",\n     \"description\": \"The location of the event\",\n
      }\n     ],\n     \"API name\": \"create_calendar_event\",\n     \"
    domain name\": \"Travel\",\n     \"parent tool name\": \"Calendar\"\n
     },\n  {\n     \"tool name\": \"Restaurant:
    get_all_restaurants_in_city\",\n     \"tool description\": \"List
    restaurant names in a city.\",\n     \"required_parameters\": [\n
    {\n     \"name\": \"city\",\n     \"type\": \"str\",\n     \"default\":
    \"San Jose\",\n     \"description\": \"The city of the event\",\n
    }\n     ],\n     \"optional_parameters\": [],\n     \"API name\": \"
    get_all_restaurants_in_city\",\n     \"domain name\": \"Travel\",\n
      \"parent tool name\": \"Restaurant\"\n  },\n  {\n     \"tool name
    \": \"Restaurant: check_restaurant_opening_hours\",\n     \"tool
    description\": \"Return operating hours; use to verify time.\",\n
    \"required_parameters\": [\n     {\n     \"name\": \"restaurant_names
    \",\n     \"type\": \"list<str>\",\n     \"default\": [\"restaurant1\",
     \"restaurant2\"],\n     \"description\": \"The names of the
    restaurants\",\n      }\n     ],\n     \"optional_parameters\": [],\n
    \"API name\": \"check_restaurant_opening_hours\",\n     \"domain name
    \": \"Travel\",\n     \"parent tool name\": \"Restaurant\"\n  },\n
    ...\n]
thought: Let's think step by step. First I need to find a vegan
    restuarant in San Jose. I need to call tool Restaurant:
    get_all_restaurants_in_city with city=San Jose. After I identify the
    vegan restaurant, I need to check if the restaurant is open that time
    . I need to call tool Restaurant: check_restaurant_opening_hours with
     restaurant_names=[\"vegan restaurant\"]. Finally, I need to set up
    this dinner on my calendar and send to Alice. I need to call tool
    Calendar: create_calendar_event with title=\"Dinner with Alice\",
    start_time=\"2025-8-19 19:00\", end_time=\"2025-8-19 21:00\",
    participants=[\"alice@example.com\"], location=\"San Jose\".
tool list: [\n{\"tool name\":\"Restaurant: get_all_restaurants_in_city\",
    \"tool description\": \"List restaurant names in a city.\", \"
    required_parameters\":[{\"name\": \"city\", \"value\": \"San Jose
    \"}], \"optional_parameters\":[], \"API name\":\"
    get_all_restaurants_in_city\", \"domain name\":\"Travel\", \"parent
    tool name\":\"Restaurant\"},\n...\n]
...
```

For the models, we use their default/recommended temperatures and allow thinking if it is a reasoning model like Claude/Gemini/...

**Retrieval-base selection**. By default, we retrieve 20 tools per time, and feed them to the model as the context.

**Details of Usage**. We use the metric Usage to measure if a tool is correctly used, i.e whether the parameters are correct. We perform direct matching against ground-truth arguments. Each tool call is represented as a JSON dictionary of the form $\{parameter\_name : parameter\_value\}$. A

predicted tool call is counted as correct only if the entire dictionary exactly matches the ground-truth dictionary after normalization. Before comparison, we apply parameter-specific normalization rules (e.g., canonicalizing date formats, removing whitespace, lowercasing strings, resolving numerical formatting) so that semantically identical values are treated consistently.

**Details of human inspections**. Below we clarify the nature, scope, and consistency of the human checks performed during dataset construction.

- Nature of human validation. As described in Section 3.2, TRAJECT-Bench is built through a multi-stage process involving tool-aware trajectory generation, real tool execution, LLM-based self-consistency query generation and refinement, and human verification. The role of human validation is to confirm that: i) each generated tool trajectory is valid and executable, ii) the final results produced by the trajectory are correct, and iii) the queries (simple and hard versions) faithfully reflect all tools, and parameters in the trajectory. All manual checks follow a fixed and objective rule: every tool and parameter in the trajectory must be correctly encoded in the query.

- Scale and extent of corrections. Because the trajectory generation includes real tool execution, all generated trajectories are already guaranteed to be valid, executable, and semantically coherent. In our inspection, none of the trajectories required manual correction. For queries, we use a self-consistency scheme to automatically align them with their trajectories: the most aligned queries are chose from multiple LLM-generated candidates, and further refined. As a result, only a very small fraction of queries required minor manual adjustments (e.g., clarifying ambiguous phrasing). These corrections were lightweight, with no need for restructuring or regeneration.

- Consistency of the validation process. Human inspection is performed with clear, deterministic rules, ensuring consistency across inspectors: i) verify that all tools appearing in the trajectory are mentioned or implied in the query; ii) confirm that parameters in the trajectory (IDs, locations, dates, quantities, etc.) match those in the query, iii) confirm that the implied user intent matches the final tool output. Because these checks are objective and grounded in the tool specifications, consistency across different inspectors is naturally maintained. We emphasize that human validation acts as a final verification layer, not the primary filtering mechanism.

## C  ADDITIONAL EXPERIMENTS

**Additional CoT results.** We present additional results of evaluating individual models with CoT prompting in Table 10 and 11. It is clear that CoT does not bring much improvement compared with the direct prompting. This may be due to the fact that almost all the evaluated models are reasoning models and we already allow the thinking mode by default. This suggest that simple direct prompting is already good enough, which is consistent with conclusions in (Guo et al., 2025).

Table 10: Results of individual LLMs using CoT on parallel queries.

| Model | Simple | | | | | Hard | | | | |
|---|---|---|---|---|---|---|---|---|---|---|
| | EM | Inclusion | Usage | Traj-Satisfy | Acc | EM | Inclusion | Usage | Traj-Satisfy | Acc |
| Claude-3.7 | 0.635 | 0.663 | 0.698 | 6.318 | 0.709 | 0.106 | 0.383 | 0.603 | 1.163 | 0.243 |
| Claude-4 | 0.826 | 0.839 | 0.799 | 8.218 | 0.882 | 0.438 | 0.636 | 0.801 | 4.460 | 0.492 |
| Gemini-2.5-flash | 0.698 | 0.719 | 0.813 | 6.945 | 0.768 | 0.203 | 0.491 | 0.692 | 2.227 | 0.250 |
| Gemini-2.5-pro | 0.774 | 0.793 | 0.735 | 7.831 | 0.867 | 0.393 | 0.628 | 0.695 | 4.119 | 0.451 |
| GPT5-mini | 0.782 | 0.801 | 0.804 | 7.911 | 0.795 | 0.168 | 0.524 | 0.681 | 1.843 | 0.369 |
| o4-mini | 0.771 | 0.762 | 0.821 | 7.671 | 0.842 | 0.344 | 0.614 | 0.732 | 3.774 | 0.464 |
| gpt-oss:120b | 0.629 | 0.645 | 0.712 | 6.358 | 0.713 | 0.158 | 0.523 | 0.657 | 1.734 | 0.355 |
| DeepSeek | 0.832 | 0.839 | 0.818 | 8.278 | 0.875 | 0.425 | 0.661 | 0.762 | 4.663 | 0.431 |
| qwen3-235b-A22B | 0.807 | 0.815 | 0.789 | 8.029 | 0.863 | 0.364 | 0.651 | 0.813 | 3.994 | 0.463 |
| Kimi-k2 | 0.783 | 0.838 | 0.869 | 7.891 | 0.875 | 0.328 | 0.642 | 0.757 | 3.599 | 0.416 |

**Additional complex structures of tool trajectories**. In the main paper, we mainly consider two basic structures, parallel and sequential. They represent the two fundamental and irreducible building blocks of tool-use behaviors. More complex graph topologies are naturally composed of these basic structures. Studying the core patterns first provides a clear foundation for understanding the

Table 11: Results of individual LLMs using CoT on sequential queries.

| Model | EM | Inclusion | Usage | Traj-Satisfy | Acc |
|---|---|---|---|---|---|
| Claude-3.7 | 0.570 | 0.650 | 0.615 | 5.950 | 0.612 |
| Claude-4 | 0.805 | 0.811 | 0.767 | 8.103 | 0.818 |
| Gemini-2.5-flash | 0.598 | 0.691 | 0.693 | 6.023 | 0.629 |
| Gemini-2.5-pro | 0.786 | 0.792 | 0.786 | 7.920 | 0.825 |
| GPT5-mini | 0.687 | 0.711 | 0.704 | 6.913 | 0.648 |
| o4-mini | 0.764 | 0.793 | 0.754 | 7.975 | 0.774 |
| gpt-oss:120b | 0.497 | 0.593 | 0.679 | 5.088 | 0.625 |
| DeepSeek | 0.818 | 0.836 | 0.805 | 8.239 | 0.798 |
| qwen3-235b-A22B | 0.786 | 0.805 | 0.784 | 7.905 | 0.757 |
| Kimi-k2 | 0.793 | 0.814 | 0.791 | 8.078 | 0.825 |

capabilities and failure modes of current models before moving to higher-order cases. In order to show the extensiveness and generalization of `TRAJECT-Bench`, we present a mixed structure of sequential and parallel. We take the Travel domain for illustration. Specifically, we consider the trajectories consisting of sequential steps, and each step may require multiple independent tool calls. Following the data generation strategy in Section 3.2, we generate the tool-calling trajectories and corresponding queries based on the tool graph and task types. We consider the sequential length (number of sequential steps) from 3 to 7, and the number of parallel tool calls from 2 to 3 for one step. We finally obtained 200 samples. We test with Claude-4, Gemini-2.5-pro, GPT5-mini, DeepSeek, Kimi-k2, following the same evaluation pipeline in Section 4.2, and we report results in Table 12. According to the results, we have similar observations in Section 4.2. Most of the tested models can achieve more than 70% EM and Acc, but merely over 80%. Moreover, the performances are worse than those in both sequential and parallel (simple version) scenarios, indicating an additional complexity from the hybrid structure. Therefore, LLMs still need improvement in utilizing tools to solve complex problems.

We will keep exploring richer structures (e.g., trees, DAGs) on diverse domains, and extend `TRAJECT-Bench`.

Table 12: Experimental results on the hybrid structure of parallel and sequential.

| Model | EM | Inclusion | Usage | Traj-Satisfy | Acc |
|---|---|---|---|---|---|
| Claude-4 | 0.710 | 0.724 | 0.745 | 6.906 | 0.690 |
| Gemini-2.5-pro | 0.705 | 0.714 | 0.756 | 6.947 | 0.695 |
| GPT5-mini | 0.555 | 0.622 | 0.650 | 5.385 | 0.505 |
| DeepSeek | 0.695 | 0.739 | 0.778 | 7.052 | 0.715 |
| Kimi-k2 | 0.700 | 0.729 | 0.762 | 7.104 | 0.720 |

**Additional agents**. In the main paper, we evaluate ReAct agent, which is widely recognized as the core agentic pattern underlying most reasoning–action loops. Therefore, including ReAct provides a strong and representative baseline. We also notice that agent architectures are large and diverse, but to show the generality and extensibility of our benchmark, we added evaluations on three additional representative agents:

* Reflexion, as mentioned by the reviewer;

* Planner–Executor (single-executor): the most popular multi-agent structure in real-world applications;

* Planner–Executor (multi-executor) with 5 clustered tool groups based on tool descriptions.

The Planner–Executor structure is particularly important because it is one of the most widely used multi-agent frameworks in practical systems (enterprise assistants, vertical-domain agents, multi-agent orchestration). As an illustration, we evaluated these agents on the Travel domain across two Claude models (Claude-3.7 and Claude-4). Results are shown in Table 13. Across both models, all three additional agents consistently outperform ReAct, indicating that `TRAJECT-Bench` is capable of differentiating agentic reasoning capabilities and capturing improvements from more advanced

frameworks. We also notice that the performance of agents on hard queries are still not satisfactory, highlighting a need for improvement. These results demonstrate that `TRAJECT-Bench`: is not restricted to single models; it is compatible with a range of agent paradigms; and can capture meaningful performance differences induced by agent architecture. Thus, our benchmark serves as a general evaluation tool that measures both model capability and agent design effectiveness. In the future, we will continue to extend the evaluation to incorporate more agents.

Table 13: Results on additional agents

| Travel Domain | | Simple | | | | | Hard | | | | |
|---|---|---|---|---|---|---|---|---|---|---|---|
| model | Agents | EM | Inclusion | Usage | Traj-Satisfy | Acc | EM | Inclusion | Usage | Traj-Satisfy | Acc |
| Claude-3.7 | ReAct | 0.762 | 0.831 | 0.773 | 7.911 | 0.801 | 0.213 | 0.619 | 0.684 | 2.157 | 0.312 |
| | Reflexion | 0.791 | 0.855 | 0.804 | 8.145 | 0.835 | 0.237 | 0.629 | 0.695 | 2.438 | 0.353 |
| | planner-executor(single) | 0.783 | 0.837 | 0.792 | 8.029 | 0.826 | 0.253 | 0.625 | 0.698 | 2.475 | 0.358 |
| | planner-executor(multi) | 0.802 | 0.869 | 0.815 | 8.282 | 0.859 | 0.264 | 0.647 | 0.717 | 2.645 | 0.383 |
| Claude-4 | ReAct | 0.901 | 0.918 | 0.868 | 9.260 | 0.931 | 0.472 | 0.681 | 0.798 | 4.901 | 0.436 |
| | Reflexion | 0.917 | 0.929 | 0.879 | 9.386 | 0.944 | 0.480 | 0.693 | 0.809 | 5.058 | 0.450 |
| | planner-executor(single) | 0.909 | 0.920 | 0.896 | 9.441 | 0.949 | 0.476 | 0.697 | 0.805 | 4.988 | 0.448 |
| | planner-executor(multi) | 0.924 | 0.935 | 0.914 | 9.581 | 0.963 | 0.484 | 0.718 | 0.819 | 5.261 | 0.462 |

**Experiments on self-correction**. Our findings show that while agent frameworks like ReAct exhibit some self-correction during tool use, this ability is limited and inconsistent, and TRAJECT-Bench helps reveal when and why such correction succeeds or fails. Because ReAct receives intermediate tool outputs and can append new reasoning steps, it sometimes corrects earlier mistakes—for example, selecting the wrong tool or adjusting an incorrect parameter. This behavior is visible in the trajectories, where the agent revises its plan after observing execution errors. This partially explains why ReAct performs better than individual modes (Table 6 vs. Table 2). However, self-correction is weak and inconsistent without explicit design. In our experiments, we do not explicitly prompt the agent to conduct self-correction, so this behavior is not consistent, and this also explains that the improvement of ReAct is marginal. To further verify, we add an additional experiment where we add an explicit instruction to encourage self-correction: you can revisit your actions and correct if any errors happen. We test on Claude models and Travel domain for illustration. The results are in Table 14. According to the results, we observe that the explicit instruction indeed encourages more self-correction behaviors and improves the performance. However, we also notice that the performance on hard queries is still not satisfactory, indicating that the self-correction is not effective for implicit intent inference, and other strategies like training or different agent architectures may be needed. `TRAJECT-Bench`provides a structured setting to measure and analyze self-correction. By providing full trajectories, intermediate outputs, and step-level correctness signals, `TRAJECT-Bench`allows researchers to observe when agents: detect their own mistakes; attempt repairs; fail to recover, ans etc. We view analyzing self-correction behaviors, under both simple and hard queries, as a key future direction, and TRAJECT-Bench establishes the foundation for systematic research in this area.

Table 14: Results on self-correction

| | | Simple | | | | | Hard | | | | |
|---|---|---|---|---|---|---|---|---|---|---|---|
| model | Agents | EM | Inclusion | Usage | Traj-Satisfy | Acc | EM | Inclusion | Usage | Traj-Satisfy | Acc |
| Claude-3.7 | ReAct | 0.762 | 0.831 | 0.773 | 7.911 | 0.801 | 0.213 | 0.619 | 0.684 | 2.157 | 0.312 |
| | ReAct+self-correction | 0.785 | 0.860 | 0.801 | 8.156 | 0.815 | 0.227 | 0.624 | 0.705 | 2.426 | 0.319 |
| Claude-4 | ReAct | 0.901 | 0.918 | 0.868 | 9.260 | 0.931 | 0.472 | 0.681 | 0.798 | 4.901 | 0.436 |
| | ReAct+self-correction | 0.922 | 0.928 | 0.893 | 9.382 | 0.946 | 0.481 | 0.695 | 0.803 | 4.973 | 0.442 |

**Experiment on LLM judges**. While we use Claude to conduct evaluation, we do not require the use of advanced models. This is because these two evaluations are not complex tasks. Traj-Satisfy mainly checks if the task is (partially) solved, and the Acc is to check if the predicted answer is equivalent to the ground truth ones. Therefore, there is a strong requirement for the model's capability. To support this, we provide additional experiments using smaller and open-source models. Specifically, we adopt GPT-oss-20 B and qwen3-8B as judges to evaluate the performance of two other models (Gemini-2.5-pro and Deepseek). We compare with Claude-4 as a judge in Table 15. It is obvious that the two small models can still effectively evaluate these metrics. Besides, we provide other metrics like EM, inclusion, which do not rely on LLMs, and provide fine-grained results.

Table 15: Results using different LLM judges.

| | Gemini-2.5-pro | | | | | |
| --- | --- | --- | --- | --- | --- | --- |
| | Parallel (simple) | | Parallel (hard) | | Sequential | |
| Judge model | Traj-Satisfy | Acc | Traj-Satisfy | Acc | Traj-Satisfy | Acc |
| Claude-4 | 8.599 | 0.911 | 4.849 | 0.498 | 8.119 | 0.848 |
| gpt-oss-20B | 8.672 | 0.903 | 4.794 | 0.482 | 8.123 | 0.849 |
| qwen3-8B | 8.677 | 0.915 | 4.859 | 0.486 | 8.174 | 0.851 |
| | Deepseek | | | | | |
| | Parallel (simple) | | Parallel (hard) | | Sequential | |
| Judge model | Traj-Satisfy | Acc | Traj-Satisfy | Acc | Traj-Satisfy | Acc |
| Claude-4 | 8.417 | 0.889 | 4.817 | 0.458 | 8.305 | 0.823 |
| gpt-oss-20B | 8.483 | 0.892 | 4.932 | 0.464 | 8.317 | 0.819 |
| qwen3-8B | 8.434 | 0.886 | 4.849 | 0.468 | 8.298 | 0.831 |

## D  FAILED EXAMPLES

We present examples corresponding to failure patterns discussed in section 4.2.

*Similar tool confusion.*

```
Query: Could you help me collect: hotel amenity codes list (limit 100);
    us states gas price; Airbnb stays near 25.7617,-80.1918 within ˜10 km
    ?
Incorrect tool list:['Airbnb listings: Listings by lat lng', 'Gas Price:
    stateUsaPrice', 'Airbnb listings: Amenities']
Correct tool list: ['Priceline com Provider: Download filter amenities',
    'Gas Price: stateUsaPrice', 'Airbnb listings: Listings by lat lng']
```

The model wrongly select 'Airbnb listings: Amenities' (mixed with 'Priceline com Provider: Download filter amenities') while the query asks for hotel amenity.

```
Query: Share the current weather there in metric, a short-term nowcast
    for that spot, and a quick read on recent air pollution?
Incorrect tool list:['AI Weather by Meteosource: current', 'Foreca
    Weather: Nowcast', 'RapidWeather: Current air pollution data']
Correct tool list: ['AI Weather by Meteosource: current', 'Foreca Weather
    : Nowcast', 'RapidWeather: Historical air pollution data']
```

The model wrongly select 'RapidWeather: Current air pollution data' (mixed with 'RapidWeather: Historical air pollution data') while the intent is to retrieve historical data.

*Parameter-blind tool selection and use.*

```
Query: ... followed by looking up airports in France, ...
Incorrect tool:{'tool name': 'Flight Data_v2: Airport data in json format
    ', 'tool description': 'Returns a file containing a comprehensive
    list of airports from the database. This endpoint is part of the
    Travelpayouts Data API, which provides valuable travel insights for
    websites and blogs by offering access to flight price trends and
    popular destination data that can help you better serve your
    customers with relevant travel information.', 'required parameters':
     [], 'optional parameters': [],...}
Correct tool: {'tool name': 'Flightera Flight Data: airportSearch',  '
    tool description': "Retrieves a list of airports based on country,
    bounding box, or timezone, allowing multiple parameters for precise
    filtering. This endpoint is part of Flightera's comprehensive service
     for flight status, on-time performance, and statistics, leveraging a
     database of 60k airports to ensure accurate and actionable data for
    flight planning and real-time analytics.",  'required parameters':
    [],  'optional parameters': [{'name': 'country', 'value': 'FR'}],...}
```

The model wrongly select a tool, which can retrieve airport data but does not have correct parameter (France).

*Redundant tool calling.*

Related but not helpful.

```
Query: What travel information is available for Switzerland? I'm
    interested in comparing options in Zurich and Geneva, including hotel
     availability and airport information.
Incorrect tool:{'tool name': 'iata_airport_codes: Get All iata airport
    codes', 'tool description': 'Retrieves a comprehensive list of all
    IATA airport codes, sourced from a globally maintained database. This
     endpoint provides access to the core dataset of the parent tool,
    which serves as an authoritative reference for airport code
    information, enabling applications to validate locations, integrate
    travel data, or analyze aviation networks with accurate, standardized
     identifiers.', 'required parameters': [], 'optional parameters':
    []}
```

Unrelated tools.

```
Query: I am planning a trip to Switzerland. I'm interested in comparing
    options in Zurich and Geneva, including hotel availability and
    airport information,...
Incorrect tool:{'tool name': 'SBB Suisse railway: Autocomplete', 'tool
    description': "Searches for train and public transport stations in
    Switzerland and across Europe, integrated with the SBB's railway
    network and public transport data to provide detailed journey
    planning, including fare calculations, departure and arrival times,
    and route options between any two stops in Switzerland.", 'required
    parameters': [{'name': 'query', 'value': 'Geneva'}], 'optional
    parameters': []}
```

*Fail to infer underlying requirements*

```
Query: I am planning a trip with my family. I'd prefer somewhere that
    past guests have loved ...
Incorrect tool:{'tool name': 'Priceline com Provider: Search hotels
    locations', 'tool description': "Searches for travel locations by
    name, allowing you to find specific destinations for your trip
    planning. This endpoint is part of priceline.com's comprehensive
    travel booking service that lets you search for hotels, rental cars,
    and flights across multiple destinations. The service provides access
     to priceline.com's extensive database of travel options, helping you
     quickly locate and compare travel destinations to build your ideal
    itinerary.", 'required parameters': [{'name': 'name', 'value': '
    Vancouver'}, {'name': 'search_type', 'value': 'ALL'}], 'optional
    parameters': [],...}
Correct tool: {'tool name': 'Hotels com Provider: Hotels Search', 'tool
    description': 'Searches for hotels based on location, dates, and
    other criteria, providing comprehensive information about available
    accommodations including prices, amenities, and reviews.', '
    required_parameters': [{'name': 'checkin_date', 'value':
    '2024-07-01'}, {'name': 'checkout_date', 'value': '2024-07-08'}, {'
    name': 'sort_order', 'value': 'REVIEW'}
```

The model fails to interpret 'somewhere that past guests have loved' into 'somewhere with good customer reviews', and therefore does not incorporate such parameters.

