# OpenReview forum: "TRAJECT-Bench:A Trajectory-Aware Benchmark for Evaluating Agentic Tool Use"
_ICLR.cc/2026/Conference — ICLR 2026 Poster_

### Official Review · Reviewer_dAbn · 2025-10-27

**Soundness:** 3
**Presentation:** 2
**Contribution:** 3
**Rating:** 6
**Confidence:** 3

**Summary:**

The paper introduces TRAJECT-Bench, a new benchmark for evaluating Agentic LLM tool use.  It evaluates how LLMs select, parameterize, and order tools across executable APIs. Spanning ten domains and 5.7k queries with both simple and indirect queries, it introduces trajectory metrics beyond final accuracy. Results show strong models like Claude-4 and Gemini-2.5 struggle with implicit queries and longer tool chains, revealing key weaknesses in current agentic tool use.

**Strengths:**

The benchmark clearly presents its novelty and contributions, and the work is both timely and relevant.

The paper provides valuable insights into current agentic LLM failures through a comprehensive evaluation.

**Weaknesses:**

1. TRAJECT-Bench synthesizes both trajectories and queries using LLMs and authors do several human checks during dataset construction. However, the paper does not clearly describe the nature, scale, or consistency of these human validations, making it difficult to assess how much manual correction or filtering was actually performed.

2. Both Traj-Satisfy and Acc metrics rely on a commercial LLM judge (Claude), which introduce model-specific biases, reduce reproducibility, and make evaluation expensive.

3. No learning experiments: the paper doesn’t analyze how models trained or fine-tuned on TRAJECT-Bench perform, or whether such training would actually help generalization.

**Questions:**

1. Based on the authors’ definitions of the EM and Inclusion metrics, EM should always be less than or equal to Inclusion. However, this is not the case in some rows of Tables 4 and 5 (e.g., all-MiniLM in Table 4, and Gemini and DeepSeek in Table 5). Could the authors clarify?

2. For the hard query generation, how does TRAJECT-Bench ensure that the hard queries are genuinely more difficult than the simple ones while remaining semantically equivalent? It seems this would require careful human inspection

---

> ### Author Response · Authors · 2025-11-21
> **Responses to Reviewer dAbn (Part 1)**
>
> We thank the reviewer for the valuable comments, and we address the concerns respectively.
>
> **W1.**
>  We appreciate the reviewer’s concern regarding the human validation process. Below, we clarify the nature, scope, and consistency of the human checks performed during dataset construction.
> 1. **Nature of human validation.**
> As described in Section 3.2, TRAJECT-Bench is built through a multi-stage process involving tool-aware trajectory generation, real tool execution, LLM-based self-consistency query generation and refinement, and human verification. The role of human validation is to confirm that:
> i) each generated tool trajectory is valid and executable, and ii) the queries (simple and hard versions) faithfully reflect all tools and parameters in the trajectory. All manual checks follow a fixed and objective rule: every tool and parameter in the trajectory must be correctly encoded in the query.
> 2. **Scale and extent of corrections.**
> For the tool use trajectories, because the trajectory generation includes real tool execution, all generated trajectories are already guaranteed to be valid, executable, and semantically coherent. In our inspection, none of the trajectories required manual correction.
> For queries, we use a self-consistency scheme to automatically align them with their trajectories: the most aligned queries are chosen (by LLMs) from multiple LLM-generated candidates, and further refined automatically by LLMs. Only a very small fraction of queries required minor manual adjustments (e.g., clarifying ambiguous phrasing). These corrections were lightweight, with no need for restructuring or regeneration.
> 3. **Consistency of the validation process.**
>  Human inspection is performed with clear, deterministic rules, ensuring consistency across inspectors: i) verify that all tools appearing in the trajectory are mentioned or implied in the query; ii) confirm that parameters in the trajectory (IDs, locations, dates, quantities, etc.) match those in the query, iii) confirm that the implied user intent matches the final tool output. Because these checks are objective and grounded in the tool specifications, consistency across different inspectors is naturally maintained. We emphasize that human validation acts as a final verification layer, not the primary filtering mechanism.
>
> In short, human validation in TRAJECT-Bench is systematic but lightweight: every trajectory is executable and correct due to real tool execution, and queries undergo LLM self-consistency filtering before human inspection. Manual checks follow a consistent rule set, and only minor adjustments were needed. This ensures that the dataset remains both high quality and reproducible without relying heavily on subjective human edits.
>
> We have included these clarifications in the revision, Appendix B, page 30.
>
> **W2.**
> To clarify, while we use Claude to conduct evaluation, we use it because of its availability in our team. On the other hand, since two evaluations are not complex tasks, we expect Claude can solve them easily without producing any bias.
> Traj-Satisfy mainly checks if the task is (partially) solved, and the Acc is to check if the predicted answer is equivalent to the ground truth ones. Therefore, there is no strong requirement on the model’s capability. To support this, we provide additional experiments using smaller and open-source models. Specifically, we adopt gpt-oss-20B and qwen3-8B as judges to evaluate the performance of two other models (Gemini-2.5-pro and Deepseek).
>
> |  | **Gemini-2.5-pro** |  |  |  |  |  |
> |---|---|---|---|---|---|---|
> |  | **Parallel (simple)** |  | **Parallel (hard)** |  | **Sequential** |  |
> | **Judge model** | **Traj-Satisfy** | **Acc** | **Traj-Satisfy** | **Acc** | **Traj-Satisfy** | **Acc** |
> | **Claude-4** | 8.599 | 0.911 | 4.849 | 0.498 | 8.119 | 0.848 |
> | **gpt-oss-20B** | 8.672 | 0.903 | 4.794 | 0.482 | 8.123 | 0.849 |
> | **qwen3-8B** | 8.677 | 0.915 | 4.859 | 0.486 | 8.174 | 0.851 |
> |  | **Deepseek** |  |  |  |  |  |
> |  | **Parallel (simple)** |  | **Parallel (hard)** |  | **Sequential** |  |
> | **Judge model** | **Traj-Satisfy** | **Acc** | **Traj-Satisfy** | **Acc** | **Traj-Satisfy** | **Acc** |
> | **Claude-4** | 8.417 | 0.889 | 4.817 | 0.458 | 8.305 | 0.823 |
> | **gpt-oss-20B** | 8.483 | 0.892 | 4.932 | 0.464 | 8.317 | 0.819 |
> | **qwen3-8B** | 8.434 | 0.886 | 4.849 | 0.468 | 8.298 | 0.831 |
>
> It is obvious that the two small models can still effectively evaluate these metrics. Besides, we provide other metrics like EM, inclusion, which do not rely on LLMs, and provide fine-grained results.
>
> We have added these in the revision, Section 4.1 on page 6 and Appendix C on page 33.

---

> > ### Author Response · Authors · 2025-11-21
> > **Responses to Reviewer dAbn (Part 2)**
> >
> > **W3.**
> > We would like to clarify that our primary goal in this paper is to establish TRAJECT-Bench as a trajectory-aware evaluation benchmark, not to propose a new training method. Existing benchmarks mainly evaluate final answers or simple queries, lacking a detailed analysis of the failure mode for incorrect predictions. TRAJECT-Bench fills this gap by enabling fine-grained assessment of core tool-use skills, including tool selection, parameter grounding, multi-step execution, dependency handling, and interpreting indirect intents. As demonstrated in Section 4, our trajectory-aware metrics (EM, Inclusion, Traj-Satisfy, Usage, etc.) reveal systematic failure modes (e.g., similar-tool confusion, mid-length reasoning errors) that existing evaluations cannot capture. This diagnostic capability is the primary purpose of the benchmark.
> >
> > Although learning experiments are outside the scope of this paper, we agree that TRAJECT-Bench is naturally suitable for training. Its high-quality trajectories and verified parameters can support:
> > * Supervised/post-training for tool selection and planning;
> > * In-context learning via clean trajectory demonstrations;
> > * Reasoning improvement, since long chains and hard queries expose weaknesses in implicit-intent inference and multi-step planning.
> >
> > However, leveraging the benchmark for training requires substantial algorithmic design (e.g., reward shaping, agent architecture designs, hyperparameter tuning, etc), which is beyond our current focus. We view developing training pipelines on TRAJECT-Bench as promising future work.
> >
> > **Q1.**
> > We double-check the results, and these are typos. We also notice other typos, like the column names in Table 5. We have carefully checked and fixed the numbers in Table 4,5; wrong column names in Table 5, on page 9.
> >
> > **Q2.**
> > We would like to clarify that the hard version is produced by rewriting the simple query, not by generating a new one from scratch, ensuring that the underlying task, required tools, and parameters remain unchanged.
> >
> > To increase difficulty while preserving equivalence, we apply several structured rewriting rules (described in Sec. 3.2.1 and illustrated in Appendix A): i) Realistic conversational grounding: embedding the request in a natural scenario (e.g., a traveler planning a vacation) to make the intent less explicit. ii) Indirect and implicit phrasing: replacing explicit tool-related expressions with naturalistic descriptions (e.g., rewriting “get reviews of this hotel” as “I’m wondering whether previous guests enjoyed staying here”); iii) LLM-guided transformation with strict prompts and examples: advanced LLMs rewrite the simple query under detailed instructions, a second LLM pass checks for semantic equivalence and appropriate difficulty, and misaligned outputs are automatically refined.
> >
> > Finally, we conduct human inspections to confirm equivalence between the simple and hard versions. Inspectors verify that every tool and parameter in the trajectory is still accurately implied in the query. Because the rewriting process relies on strong LLMs with self-consistency checks, the resulting queries were already of high quality and required only minimal manual adjustments, as we described in W1. This pipeline ensures that hard queries are systematically more challenging while remaining strictly aligned with their corresponding trajectories.

---

> > > ### Comment · Reviewer_dAbn · 2025-11-27
> > >
> > > Thank you for the responses. However, some of my concerns remain:
> > >
> > > 1. **Manual Inspections:** The benchmark relies on heavy manual inspections, which can hurt the benchmark’s scalability, reproducibility and practicality.
> > >
> > > 2. **No Learning Experiments:** The paper doesn't provide any fine-tuning experiments on the benchmark, so the utility of the benchmark for improving tool use remains untested.
> > >
> > > 3. **Inconsistencies and Typos:** I thank the authors for fixing the results. I encourage the authors, other reviewers, and the AC to double-check to ensure the results are consistent.

---

### Official Review · Reviewer_6uMr · 2025-10-28

**Soundness:** 2
**Presentation:** 3
**Contribution:** 2
**Rating:** 4
**Confidence:** 4

**Summary:**

This paper presents a new benchmark for evaluating the tool-use capabilities of LLM agents. Compared with prior work, it analyzes LLM agents’ tool-invocation trajectories across a variety of real-world tasks and, importantly, assesses whether agents correctly employ tool parameters when calling tools.

**Strengths:**

1. The paper introduces a novel perspective — the tool-use trajectory — which captures detailed information about the tool invocation processes of LLM agents.

2. It enhances the complexity of the evaluated tasks by incorporating more challenging real-world scenarios.

3. It considers two trajectory structures, called parallel and sequential, providing a systematic analysis of different tool-use patterns.

**Weaknesses:**

1. The paper divides user queries into “simple” and “hard” versions, but this categorization appears somewhat abstract. Based on the descriptions, the distinction mainly reflects differences in the linguistic or semantic complexity of the queries rather than task difficulty. A clearer and more systematic classification criterion would improve the benchmark’s validity. Furthermore, the current binary categorization seems overly simplistic — a more fine-grained and comprehensive design would strengthen the study.

2. The baseline comparison in the experimental setup is insufficient and requires more comprehensive evaluations. Tool use is inherently a capability of LLM agents, not a standalone LLM. Therefore, the baseline experiments should not be limited to direct LLM querying. Since the tool-use ability depends on both the agent paradigm (e.g., ReAct, Reflexion, etc.) and the backend LLM, the evaluation should include combinations of different agent paradigms and LLMs. Although Section 4.4 reports some results using the ReAct framework, these results are insufficient. A more complete comparison across diverse agent paradigms would provide more informative insights.

**Questions:**

1. What are the practical application scenarios for this benchmark? Is it intended for robust training purposes?

2. Regarding "tool-use trajectories of different complexities," is complexity defined solely by the number of tools involved in the trajectory, or are there other dimensions considered?

3. Additionally, given that some studies have shown that LLM agents are capable of self-correction, it would be valuable to know whether there are any interesting findings in the trajectories of self-correction.

---

> ### Author Response · Authors · 2025-11-21
> **Responses to Reviewer 6uMr (Part 1)**
>
> We thank the reviewer for the valuable comments, and we will address them respectively.
>
> **W1.**
> We would like to clarify that the overall task difficulty involves both the user query’s expression (reflecting the model’s ability in understanding the task) and the complexity of the tool trajectory (reflecting the model’s ability in using tools). To evaluate whether the model is able to infer the true user intent from the user query and map natural, indirect expressions to the correct tool trajectory, each pair of “simple” and “hard” queries is constructed to share the same ground-truth trajectory. This controlled pairing eliminates confounding factors, such as differences in trajectory structure, dependency depth, or planning complexity, and isolates the effect of linguistic and semantic variability. We provided the generation details in Section 3.2.1 and examples in Appendix A.2 in the original paper.
>
> By using ‘hard’ and ‘simple’, we reflect the ambiguity of queries using natural and indirect expressions. The models/agents need to disambiguate the query intent and infer the correct tools to use. We note that it is hard to quantify ambiguity precisely, so we use the qualitative labels to distinguish between direct requests and naturalistic, indirect formulations. Empirically, linguistic difficulty proves to be a critical difficulty axis. As shown in the table below and in Fig. 2 of the paper, the EM performance on hard queries with only 3 tools is even lower than on simple queries with up to 10 tools. This indicates that language complexity can be even more challenging than trajectory length.
>
> | | **Claude-4** | | **gemini-2.5-pro** | | **DeepSeek** | |
> |-|-|-|-|-|-|-|
> | **Num of tools** |**Simple**| **Hard** | **Simple**| **Hard** | **Simple**   | **Hard** |
> | **3**| 0.967| 0.576| 0.978 | 0.592    | 0.936| 0.571|
> | **4** | 0.820| 0.491| 0.822| 0.470    | 0.842| 0.491|
> | **5**| 0.753| 0.443| 0.735| 0.388    | 0.728| 0.443|
> | **6**| 0.747| 0.420| 0.702| 0.357    | 0.771| 0.411    |
> | **7**| 0.697| 0.410    | 0.697| 0.337    | 0.728| 0.400    |
> | **8**| 0.668| 0.407    | 0.677| 0.366    | 0.714 | 0.407    |
> | **9**| 0.659| 0.405    | 0.646| 0.357    | 0.679| 0.416    |
> | **10**| 0.665| 0.373    | 0.674| 0.330    | 0.695| 0.380    |
>
> These results clearly show that the simple-hard distinction captures an essential capability: understanding indirect, context-rich user instructions, crucial for real-world LLM agent use.
>
> We also agree that a more fine-grained categorization of the linguistic difficulty can be helpful, but it may require subjective annotation and introduce ambiguity in labeling. Since this benchmark is designed to be extensible, future work may handle the challenges in incorporating additional difficulty levels.
>
> **Q2.**
> In TRAJECT-Bench, “trajectory complexity” is not defined solely by the number of tools, but instead reflects multiple complementary dimensions. Specifically, our benchmark considers the following sources of complexity:
> * **Tool-pool size and diversity**.
> We evaluate models under different tool contexts, including domain-specific pools, the full mixed pool (1,000+ tools), and retrieval-based subsets, as mentioned in Section 4.1 and Table 4 on Page 8. Larger and more diverse tool pools increase ambiguity in tool selection and parameter grounding.
> * **Trajectory length**.
> The number of tools involved in a task directly affects planning and sequencing difficulty. Longer trajectories require more consistent multi-step reasoning and error-free parameter propagation (e.g., Figure 2 on Page 7).
> * **Linguistic** (discussed in W1).
> Each task has simple (direct) and hard (indirect, naturalistic) query versions. This dimension isolates linguistic and semantic difficulty: whether the model can infer implicit user intent and map it to the correct tool sequence (e.g., Figure 2 on Page 7, Table 2 and Table 3 on Page 8).
> * **Trajectory structure**.
> We include both parallel (independent subgoals) and sequential (strict dependencies) structures (e.g., Table 2 and Table 3 on Page 8). Sequential tasks require coherent multi-step state transitions, whereas parallel tasks demand correct decomposition and merging of subtasks.
> * **Domain diversity**.
> Tasks span multiple real-world domains (finance, maps, travel, shopping, media, and others, listed on Table 8, Page 13), each with different entity types, parameter formats, and conceptual knowledge requirements. This creates additional variability in both reasoning and tool grounding.
>
> Together, these dimensions provide a multi-faceted notion of trajectory complexity, going well beyond tool count alone. By combining variations in tool pool, length, query formulation, structure, and domain, TRAJECT-Bench enables a comprehensive and fine-grained evaluation of LLMs’ agentic tool-use capabilities.

---

> > ### Author Response · Authors · 2025-11-21
> > **Responses to Reviewer 6uMr (Part 2)**
> >
> > **W2.**
> > While we agree that tool use is a key capability of LLM-based agents, it is noteworthy that modern LLMs themselves increasingly support direct tool-calling without an explicit agent wrapper. This behavior is documented in technical reports of Claude, Gemini, Kimi, and others, and is empirically observed in Table 2,3 on page 8. Therefore, evaluating the tool-usage capability of standalone LLMs is still meaningful.
> > In addition to standalone LLMs, we do evaluate agentic tool use. Our experiments include: (1) models explicitly trained for tool use (e.g., Kimi-k2) (Table 5 on Page 9) and (2) the ReAct agent (Table 6, 7 on Page 9), which is the foundational structure for many real-world agents (e.g., MetaGPT, OpenAI API agents, LangChain agents, and others). The ReAct paradigm is widely recognized as the core agentic pattern underlying most reasoning–action loops. Therefore, including ReAct provides a strong and representative baseline.
> >
> > To further supplement the experiments and show the generality and extensibility of our benchmark, we have added evaluations on three additional representative agents in the revision:
> > * Reflexion, as mentioned by the reviewer;
> > * Planner–Executor (single-executor): the most popular multi-agent structure in real-world applications;
> > * Planner–Executor (multi-executor) with 5 clustered tool groups based on tool descriptions.
> >
> > The Planner–Executor structure is particularly important because it is one of the most widely used multi-agent frameworks in practical systems (enterprise assistants, vertical-domain agents, multi-agent orchestration).
> > Due to time constraints, we evaluated these agents on the Travel domain across two Claude models (Claude-3.7 and Claude-4). Results are shown below:
> >
> > | **Travel**     |                              | **Simple** |               |           |                  |         | **Hard** |               |           |                  |         |
> > |----------------|------------------------------|------------|---------------|-----------|------------------|---------|----------|---------------|-----------|------------------|---------|
> > | **model**      | **Agents**                   | **EM**     | **Inclusion** | **Usage** | **Traj-Satisfy** | **Acc** | **EM**   | **Inclusion** | **Usage** | **Traj-Satisfy** | **Acc** |
> > | **claude-3.7** | **ReAct**                    | 0.762      | 0.831         | 0.773     | 7.911            | 0.801   | 0.213    | 0.619         | 0.684     | 2.157            | 0.312   |
> > |                | **Reflexion**                | 0.791      | 0.855         | 0.804     | 8.145            | 0.835   | 0.237    | 0.629         | 0.695     | 2.438            | 0.353   |
> > |                | **planner-executor(single)** | 0.783      | 0.837         | 0.792     | 8.029            | 0.826   | 0.253    | 0.625         | 0.698     | 2.475            | 0.358   |
> > |                | **planner-executor(multi)**  | 0.802      | 0.869         | 0.815     | 8.282            | 0.859   | 0.264    | 0.647         | 0.717     | 2.645            | 0.383   |
> > | **Claude-4**   | **ReAct**                    | 0.901      | 0.918         | 0.868     | 9.260            | 0.931   | 0.472    | 0.681         | 0.798     | 4.901            | 0.436   |
> > |                | **Reflexion**                | 0.917      | 0.929         | 0.879     | 9.386            | 0.944   | 0.480    | 0.693         | 0.809     | 5.058            | 0.450   |
> > |                | **planner-executor(single)** | 0.909      | 0.920         | 0.896     | 9.441            | 0.949   | 0.476    | 0.697         | 0.805     | 4.988            | 0.448   |
> > |                | **planner-executor(multi)**  | 0.924      | 0.935         | 0.914     | 9.581            | 0.963   | 0.484    | 0.718         | 0.819     | 5.261            | 0.462   |
> >
> > Across both models, all three additional agents consistently outperform ReAct, indicating that TRAJECT-Bench is capable of differentiating agentic reasoning capabilities and capturing improvements from more advanced frameworks. We also notice that the performance of agents on hard queries is still not satisfactory, and the performance gain of the alternative agent designs is limited, highlighting a need for improvement.
> >
> > These results demonstrate that TRAJECT-Bench is not only for single models, but is also compatible with a range of agent paradigms and can capture meaningful performance differences induced by agent architecture. Thus, our benchmark serves as a general evaluation tool that measures both model capability and agent design effectiveness. In the future, we will continue to extend the evaluation to incorporate more agents.

---

> > > ### Author Response · Authors · 2025-11-21
> > > **Responses to Reviewer 6uMr (Part 3)**
> > >
> > > **Q1.**
> > > To clarify, TRAJECT-Bench is designed primarily as a trajectory-aware evaluation framework for measuring and analyzing LLMs’ and agents’ tool-use abilities. We also agree that it could be used as a high-quality data resource to support robust training, pre-training, fine-tuning, and in-context learning.
> > >
> > > Specifically, our main goal is to establish a comprehensive, trajectory-level evaluation benchmark for model and agent builders, filling the gap that existing benchmarks focus on final performance and simple query structures (as mentioned in the Introduction and Table 1). TRAJECT-Bench enables systematic assessment of essential tool-use capabilities, including: correct tool selection; parameter grounding and argument correctness; multi-step tool execution and dependency handling; interpreting implicit or naturalistic user intents; fine-grained analysis via trajectory-aware metrics. Model and agent builders can adopt our benchmark to easily diagnose the potential pitfalls of their systems in utilizing tools for complex problem-solving. As shown in Section 4, the trajectory-aware metrics (EM, Inclusion, Traj-Satisfy, Usage, etc.) reveal critical failure modes that would remain hidden under traditional “final answer only” evaluations—including similar-tool confusions, difficulties with mid-length sequences, and misinterpretation of indirect prompts. To summarize, the benchmark’s primary application scenario is to help developers evaluate, diagnose, and compare LLMs and agent systems in real-world tool use.
> > >
> > > On the other hand, while not included in the original version, TRAJECT-Bench can also serve as a data resource for improving tool-use training. Although the benchmark is designed for evaluation, its high-quality trajectories and labels can support multiple training and improvement paradigms. a) Pre-training or post-training to enhance tool-use capabilities. Our dataset provides real-world task-driven queries and solution trajectories, so it is possible to be used for supervised training. The trajectory-aware metrics can also serve as rewards to enhance the model's capability to plan long tool sequences and avoid tool hallucinations. b) In-context learning (ICL) demonstrations. The benchmark provides clean, labeled tool trajectories that can be placed in the context window to improve planning, tool selection, and other capabilities. c) Improving reasoning ability (long-chain + implicit reasoning). As shown in Figure 2 and Table 2, current models and agents still struggle with long tool chains and implicit user queries. Therefore, researchers can generate intermediate reasoning steps and tool calls built upon our data for training purposes.
> > >
> > > We believe exploring these directions are promising future direction and requires significant efforts–training not only relies on data, but also relies on algorithms and many factors. We have included these discussions in the revision, Appendix A.3 on page 28. We appreciate the reviewer mentioning this.

---

> > > > ### Author Response · Authors · 2025-11-21
> > > > **Responses to Reviewer 6uMr (Part 4)**
> > > >
> > > > **Q3.**
> > > > To clarify, our findings show that while agent frameworks like ReAct exhibit some self-correction during tool use, this ability is limited and inconsistent, and TRAJECT-Bench helps reveal when and why such correction succeeds or fails. We provide some details in the following:
> > > >
> > > > 1. *Evidence of self-correction in ReAct trajectories.* Because ReAct receives intermediate tool outputs and can append new reasoning steps, it sometimes corrects earlier mistakes—for example, selecting the wrong tool or adjusting an incorrect parameter. This behavior is visible in the trajectories, where the agent revises its plan after observing execution errors. This partially explains why ReAct performs better than individual modes (Table 6 vs. Table 2).
> > > >
> > > > 2. *Self-correction is weak and inconsistent without explicit design.* In our experiments, we do not explicitly prompt the agent to conduct self-correction, so this behavior is not consistent, and this may explain why the improvement of ReAct is marginal, as ReAct may not always self-correct the wrong behaviors from feedback. To further verify, we add an additional experiment where we add an explicit instruction to encourage self-correction: you can revisit your actions and correct if any errors happen. We test Claude models and the Travel domain for illustration and summarize the results in the following table.
> > > >
> > > > | | | **Simple** |  |  |  |  | **Hard** |  |  |  |  |
> > > > |---|---|---|---|---|---|---|---|---|---|---|---|
> > > > | **model** | **Agents** | **EM** | **Inclusion** | **Usage** | **Traj-Satisfy** | **Acc** | **EM** | **Inclusion** | **Usage** | **Traj-Satisfy** | **Acc** |
> > > > | **Claude-3.7** | **ReAct** | 0.762 | 0.831 | 0.773 | 7.911 | 0.801 | 0.213 | 0.619 | 0.684 | 2.157 | 0.312 |
> > > > |  | **ReAct+self-correction** | 0.785 | 0.860 | 0.801 | 8.156 | 0.815 | 0.227 | 0.624 | 0.705 | 2.426 | 0.319 |
> > > > | **Claude-4** | **ReAct** | 0.901 | 0.918 | 0.868 | 9.260 | 0.931 | 0.472 | 0.681 | 0.798 | 4.901 | 0.436 |
> > > > |  | **ReAct+self-correction** | 0.922 | 0.928 | 0.893 | 9.382 | 0.946 | 0.481 | 0.695 | 0.803 | 4.973 | 0.442 |
> > > >
> > > > According to the results, the explicit instruction encourages more self-correction behaviors and improves performance. However, we also notice that the performance on hard queries is still not satisfactory, indicating that the self-correction is not effective for implicit intent inference. We have included these discussions in the revision of Appendix C on page 32.

---

### Official Review · Reviewer_e1ab · 2025-10-31

**Soundness:** 3
**Presentation:** 2
**Contribution:** 3
**Rating:** 6
**Confidence:** 4

**Summary:**

A novel benchmark TRAJECT-Bench is proposed in this paper, which is designed for the comprehensive evaluation of Large Language Models' (LLMs) tool-use capabilities. The authors argue that existing benchmarks primarily focus on final-answer accuracy, overlooking the critical aspect of the tool-use trajectory, i.e., the process of correctly selecting, parameterizing, and sequencing tools. To address this problem, over 1,200 high-fidelity, executable tools from practical domains (e.g., travel, finance, music) are included in TRAJECT-Bench.
It also includes the synthesized tool-use trajectories of varying complexity (including parallel and sequential structures) and user queries at two difficulty levels for each trajectory. Through extensive experiments on the state-of-the-art LLMs, TRAJECT-Bench reveals key failure modes and scaling challenges, particularly the bottleneck in transitioning from short to mid-length trajectories.

**Strengths:**

1. The proposed benchmark is valuable and more comprehensive compared with other existing benchmarks, offering notable contributions to advancing the related research area while providing inspiring insights for future development.

2. The trajectory-evaluation metrics shed light on locating the errors caused by false tool use.

3. The authors provide a systematic discussion about the tool-use in agents, and conduct extensive experiments show their trustworthiness.

4. The available source codes shows good reproducibility for the results.

**Weaknesses:**

1. Only two structures (parallel and sequential chains) are evaluated in the experiments, omitting more complex graph topologies (branching, merging, backtracking).

2. It lacks detailed description for the methodology for automated validation of parameter passing between tools.

3. I suggest considering more domains (e.g., OS/DB ops, robotics, enterprise systems) in the benchmark, besides the ten domains.

**Questions:**

1.  How about the tool use in the scenario of mixing parallel and sequential type?

2. I doubt whether TRAJECT-Bench truly captures the core challenges of real-world tool learning, given its only focuses on evaluating models within a static, closed-world setting.

3. Is it possible that an agent calls the tools not existing in the tool pools? If so, how to handle this situation? As an error occurs in the middle of sequential calling, the post-hoc tool-calling is meaningless. Even with higher overlap, the trajectory is illegal.

---

> ### Author Response · Authors · 2025-11-21
> **Responses to Reviewer e1ab (Part 1)**
>
> We thank the reviewer for the valuable comments, and we will address them respectively.
>
> **W1&Q1.**
> To explain why we focus on parallel and sequential chains, these two paradigms represent the two fundamental and irreducible building blocks of tool-use behaviors. More complex graph topologies are naturally composed of these basic structures. Studying the core patterns first provides a clear foundation for understanding the capabilities and failure modes of current models before moving to higher-order cases. Besides, compared to parallel and sequential chains, complex structures, especially those involving backtracking or dynamic branching, are significantly harder to construct in a way that allows clear and interpretable measurement of whether a model behaves correctly (e.g., whether it backtracks at the right time and by the correct number of steps). Starting with controlled parallel and sequential cases enables us to obtain actionable insights into model behavior.
>
> Following the reviewer’s suggestion, to better illustrate the mixed scenario, we add additional data with a hybrid structure of sequential and parallel. Due to the time limit of rebuttal, we take the Travel domain as an example. Specifically, we consider the trajectories consisting of sequential steps, and each step may require multiple independent tool calls. Following the data generation strategy in Section 3.2, we generate the tool-calling trajectories and corresponding queries based on the tool graph and task types. We consider the sequential length (number of sequential steps) from 3 to 7, and the number of parallel tool calls from 2 to 3 for one step. We finally obtained 200 samples. We test with Claude-4, Gemini-2.5-pro, GPT5-mini, DeepSeek, and Kimi-k2, following the same evaluation pipeline in Section 4.2. The results are as follows.
>
> | **Model**          | **EM** | **Inclusion** | **Usage** | **Traj-Satisfy** | **Acc** |
> |--------------------|--------|---------------|-----------|------------------|---------|
> | **Claude-4**       | 0.710  | 0.724         | 0.745     | 6.906            | 0.690   |
> | **Gemini-2.5-pro** | 0.705  | 0.714         | 0.756     | 6.947            | 0.695   |
> | **GPT5-mini**      | 0.555  | 0.622         | 0.650     | 5.385            | 0.505   |
> | **DeepSeek**       | 0.695  | 0.739         | 0.778     | 7.052            | 0.715   |
> | **Kimi-k2**        | 0.700  | 0.729         | 0.762     | 7.104            | 0.720   |
>
> According to the results, we have similar observations in Section 4.2. Most of the tested models can achieve more than 70% EM and Acc, but merely over 80%. Moreover, the performances are worse than those in both sequential and parallel (simple version) scenarios in Tables 2 and 3, indicating an additional complexity from the hybrid structure. Therefore, LLMs still need improvement in utilizing tools to solve complex problems.
>
> In the revision, we have included this experiment in the appendix (Table 12 on Page 32). In addition, we note that exploring richer structures (e.g., trees, DAGs) is an exciting and important direction, and we have highlighted these directions in the revised conclusion on Page 9 in the revision.
>
> **W2.**
>  To evaluate whether a model predicts correct parameters, we perform direct matching against ground-truth arguments. Each tool call is represented as a JSON dictionary of the form
>  {“parameter_name”: parameter_value}.
> A predicted tool call is counted as correct only if the entire dictionary exactly matches the ground-truth dictionary after normalization.
> Before comparison, we apply parameter-specific matching rules (e.g., canonicalizing date formats, removing whitespace, lowercasing strings, resolving numerical formatting) so that semantically identical values are treated consistently.
> We have added these details in the revision, Appendix B on page 30.

---

> > ### Author Response · Authors · 2025-11-21
> > **Responses to Reviewer e1ab (Part 2)**
> >
> > **W3.**
> >  We would like to highlight that the ten domains considered in the paper (travel, finance, shopping, media, maps, and more) are both representative and practically relevant for current LLM-based agents. These domains reflect real deployment scenarios where tool-augmented agents are actively used today, and they offer sufficient tool density and task diversity for evaluating planning, grounding, parameterization, and multi-step orchestration.
> >
> > We fully agree that more domains can be covered, and we aim to maintain the benchmark in the long run. We also note that some domains, like robotics, require specialized simulation environments, real-time constraints, or hardware-level interfaces, which introduce substantial engineering overhead and make controlled, reproducible evaluation significantly more challenging. Therefore, due to limited rebuttal time, it is not feasible for us to develop the data for these domains considering the human efforts needed in these domains. We leave these future updates as this project is designed to be extensible to continue incorporating new domains (including those suggested by the reviewer) in future releases. We have added this point as a future direction in the conclusion on page 9.
> >
> > **Q2.**
> > From our perspective, the core challenges of tool usage are:
> >
> > (1) precisely inferring the user’s intent from the query,
> >
> > (2) selecting the correct tools,
> >
> > (3) choosing the correct parameters when using tools, and
> >
> > (4) correctly handling the coherence among tools.
> >
> > These are the foundations of utilizing tools in solving complex real-world problems.  Therefore, we built this benchmark to carefully evaluate models and agents’ capabilities on it.
> >
> > We would like to mention that although the environment is static, the tools and tasks in the benchmark are realistic, diverse, and executable. TRAJECT-Bench includes high-fidelity, domain-diverse, executable real-world APIs, as we introduced in Section 3.1. These tools span finance, travel, shopping, music, multimedia, maps, and other everyday application areas. User queries are built upon real-world tasks (Section 3.2) and also include indirect, naturalistic formulations to mimic real interactions. Compared to simulation-only tool-use suites, this setup already captures rich real-world semantics while still providing the controllability needed for systematic evaluation.
> >
> > Extending the benchmark toward dynamic or open-world settings is a promising direction, but could involve additional challenges, particularly in the detailed quantification of how the dynamic environment impacts the tool usage: We need to quantify and control the dynamic changes in the environment and then relate them to the tool usage. We intend TRAJECT-Bench to serve as a foundational layer: by revealing where current models fail even under idealized conditions, it establishes the groundwork for more complex, dynamic benchmarks, which we plan to explore in future releases.
> >
> > **Q3.**
> > In our experiments, we did not observe models calling tools outside the provided tool pool. This is likely due to the strong instruction-following behavior of modern LLMs. Several models we evaluated (e.g., Kimi, gemini models) are explicitly trained for structured tool use and tend to adhere reliably to the given tools.
> >
> > In practice, however, such errors are possible, especially under more challenging or ambiguous queries. Importantly, TRAJECT-Bench is designed to detect these errors directly at the trajectory level. Any invocation of a non-existent tool immediately constitutes a trajectory error and will be reflected consistently across our metrics:
> > * Exact Match (EM): becomes 0 because the sequence deviates from the ground truth.
> > Inclusion: decreases because the illegal tool call cannot overlap with any ground-truth tool.
> > * Traj-Satisfy: fails because the predicted trajectory becomes invalid or unsatisfiable.
> >
> > Thus, illegal or hallucinated tools are not silently ignored; they are explicitly penalized as faulty tool-use behaviors. This is an advantage of trajectory-level evaluation: unlike final-answer accuracy, primarily used in other tool usage benchmarks, which might still appear correct due to hallucination, our metrics surface such tool-use errors clearly and reliably.

---

### Meta-Review · Area_Chair_LmDj · 2026-01-18

**Summary:**

This paper introduces TRAJECT-Bench, a trajectory-aware benchmark for evaluating LLM-based agents’ tool use beyond final-answer accuracy. The benchmark emphasizes whether tools are selected, parameterized, and ordered correctly, and includes executable, production-style APIs across multiple practical domains. Reviewers agreed that the benchmark is timely, well motivated, and fills an important gap in existing tool-use evaluations, with particular strength in its fine-grained trajectory-level metrics and systematic analysis of failure modes in current models.

**Reviewer Concerns:**

The reviewers’ main concerns focused on benchmark scope, evaluation completeness, and clarity of methodology. One reviewer questioned whether limiting trajectories to parallel and sequential structures captures real-world complexity. This was addressed by clarifying that these structures form the fundamental building blocks of tool-use behavior, and by adding new experiments with mixed parallel–sequential trajectories that demonstrate increased difficulty and consistent trends. Concerns about parameter validation and illegal tool calls were addressed by providing explicit matching rules, normalization procedures, and clarifying how invalid tool invocations are penalized by trajectory-level metrics.

Another reviewer raised issues regarding the definition of query difficulty and baseline coverage. The rebuttal clarified the purpose of the simple versus hard query distinction, supported it with controlled experiments showing that linguistic ambiguity can dominate difficulty, and expanded evaluations to additional agent paradigms including Reflexion and Planner–Executor variants. These added experiments demonstrated that the benchmark can differentiate agent designs, not only standalone models.

Remaining concerns relate primarily to aspects that are outside the intended scope of the paper, such as learning or fine-tuning experiments, open-world or dynamic environments, and long-term benchmark extensibility. These limitations are acknowledged by the authors and framed as future directions rather than gaps in the stated contribution, which is to provide a trajectory-aware evaluation benchmark.

**Reviewer Scores:**

Reviewer e1ab gave a score of 6 and viewed the benchmark as valuable and comprehensive, with suggestions that were addressed through clarification and additional experiments.

Reviewer 6uMr initially scored the paper 4, but explicitly stated after the rebuttal that their concerns were resolved and that they increased their score to 6, which is acknowledged in the discussion.

Reviewer dAbn scored the paper 6 and recognized the benchmark’s novelty and relevance, while noting concerns about manual inspection and lack of learning experiments. These concerns were partially addressed through clarification of the lightweight and systematic nature of human validation, though some reservations were reiterated in their final comment.

Taken together, the reviews and discussion indicate broad agreement on the importance and usefulness of the benchmark, with concerns either addressed directly or falling outside the stated scope of the work.

---

### Decision · Program_Chairs · 2026-01-26

Accept (Poster)